# A systematic review of individual, social, and societal resilience factors in response to societal challenges and crises

Sarah K. Schäfer [1,2] ✉, Max Supke[1,2], Corinna Kausmann[3], Lea M. Schaubruch[1], Klaus Lieb[1,4] & Caroline Cohrdes [3]

Societal challenges put public mental health at risk and result in a growing interest in resilience as trajectories of good mental health during stressor exposure. Resilience factors represent multilevel psychosocial resources that increase the likelihood of resilient responses. This preregistered systematic review aims at summarizing evidence on the predictive value of individual, social and societal resilience factors for resilient responses to societal challenges and crises. Eligible studies examined the predictive value of resilience factors in stressor-exposed populations in high-income countries by means of multinomial regression models based on growth mixture modeling. Five databases were searched until August 2, 2023. Data synthesis employed a rating scheme to assess the incremental predictive value of resilience factors beyond sociodemographic variables and other resilience factors. An adapted version of the Newcastle-Ottawa Scale was used for risk of bias assessment. Fifty studies (sample sizes: 360–65,818 participants) with moderate study quality reported on various stressors (e.g., pandemics, natural disasters, terrorist attacks). Higher income, socioeconomic status and perceived social support, better emotion regulation and psychological flexibility were related to more resilient responses. The association between resilience factors and resilient responses was stronger in samples with younger mean age and a larger proportion of women. Most studies used non-representative convenience samples and effects were smaller when accounting for sociodemographic variables and other resilience factors. For many factors, findings were mixed, supporting the importance of the fit between resilience factors and situational demands. Research into social and societal resilience factors and multilevel resilience interventions is needed. Preregistration-ID: 10.17605/OSF.IO/GWJVA. Funding source: Robert Koch Institute (ID: LIR_2023_01).

Within the last years, many societies were exposed to multiple stressors and crises such as the COVID-19 pandemic, economic crises, wars and armed conflicts, or natural disasters[1]. Beyond differences between those stressors, they share relevant similarities as they affect a large number of people relatively synchronously and have potentially long-lasting consequences for societies[2], leading to increased stress for many individuals[3,4]. As stress is among the leading causes for the onset and persistence of mental disorders, those societal challenges put the mental health of substantial parts of the population at risk, resulting in a serious public mental health issue[5–8]. This

poses the question of how people can maintain or regain their mental health in face of societal challenges—that is, how people can respond resiliently to stress.

While resilience has often been viewed as the personal capacity to bounce back after exposure to stress[9], recent approaches in resilience research conceptualize resilience as an outcome, that is, favorable adaptation in face of stress[10,11]. More precisely, resilience as an outcome can be defined as the maintenance or fast recovery of mental health during or after stressor exposure[11]. Within this framework, so-called resilience factors protect

[1]Leibniz Institute for Resilience Research, Mainz, Germany. [2]Department of Clinical Psychology, Psychotherapy and Psychodiagnostics, Technische Universität, Braunschweig, Braunschweig, Germany. [3]Department of Epidemiology and Health Monitoring, Robert Koch Institute, Berlin, Germany. [4]Department of Psychiatry and Psychotherapy, University Medical Center of Johannes Gutenberg University, Mainz, Germany. ✉e-mail: sarah.schaefer@lir-mainz.de

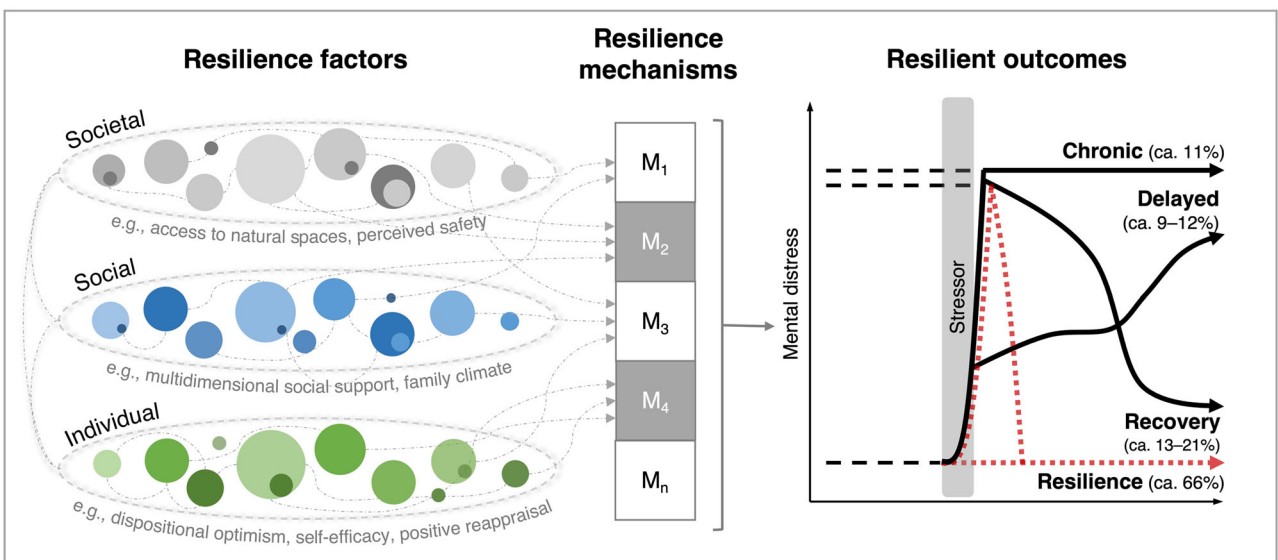

**Fig. 1 | Theoretical framework underlying this systematic review.** Illustration of the link between resilience factors and resilience outcomes mediated via higher-level resilience mechanisms. This idea is based on recent ideas in resilience research[11,22,23,33] and has been adapted for the multilevel resilience factor approach of this review. Note that also (neuro)biological and (epi)genetic factors are discussed as individual resilience factors[23,166], however, those are not focus of this review. In this figure, we present a trajectory of fast recovery after stressor exposure in light red, which is also labeled as 'resilience'. This reflects the idea that not any kind of mental response to stress is pathological per se[22]. However, we acknowledge the fact that such trajectories of very fast recovery of mental health have rarely been identified in primary studies, which might also reflect problems of timing as such responses could only be captured by high-frequency assessments[167].

individuals from potentially negative effects of stressors and increase the likelihood of resilient responses[12,13]. Following a multisystemic approach to resilience[14], resilience factors represent individual, social and societal resources. Individual resilience factors include psychological variables like dispositional optimism[15] and self-efficacy beliefs[16,17], while social resilience factors represent perceived and available resources in one's social environment such as perceived social support[18,19] and family cohesion[20]. Societal resilience factors refer to resources that are either perceived or available at a societal level, for example, resources in the built or natural environment like access to natural green or blue (e.g., parks or lakes)[21]. Due to the large number of resilience factors at different levels, recent conceptual approaches claim that a smaller number of resilience mechanisms might mediate the relationship between resilience factors and resilient outcomes[11,14,22] (see Fig. 1). For example, at the individual level, Kalisch et al.[22] proposed that many psychological resilience factors impact on resilient outcomes, with positive appraisal style (i.e., non-pessimistic, non-catastrophic, non-helpless types of appraisal[17]) being the key mediator. More recently, Kalisch et al.[23] further differentiated rather stable differences in positive appraisal style as resilience factor from positive appraisal used for coping during stressor exposure, with the latter representing a resilience mechanism. In a similar vein, Bonanno[24,25] suggested regulatory flexibility as an overarching mechanism for resilient outcomes, with regulatory flexibility reflecting one's ability to modulate emotional experiences and the perceived ability to make use of different coping strategies depending on contextual demands and feedback. So far, a small number of primary studies[26–29] provides support for these ideas; however, evidence is still rare and comprehensive tests of more complex models also including associations with resilience factors are still missing. At the social and societal level, other resilience mechanisms come into play (e.g., decision making, use of societal resources, capacities for transformation)[30,31], however, those have rarely been examined in primary studies on individual mental responses to societal challenges.

The most common approach in these studies is to examine trajectories of mental distress and, less often, positive mental health in response to stressor exposure[32,33]. Based on a hallmark paper by Bonanno et al.[34], most studies employ different types of growth mixture modeling (GMM)[35,36]. GMM aims at identifying multiple unobserved sub-populations, which show specific patterns of change over time (see Supplementary Note 1 for

details). Recent reviews[17,33] on studies employing this method to examine responses to major stressors showed that approximately 66% of stress-exposed individuals respond resiliently, that is, show a trajectory of stable low mental distress or good mental health (*resilience*). Another 13–21% show *recovery* responses (also emergent resilience[37]), that is, initial increases in mental distress followed by later decreases. A pattern of initially low mental distress and later increases in mental distress is shown by 9–12% of stress-exposed individuals (*delayed*) and approximately 11% are found to report consistently high levels of mental distress (*chronic*). Among the most common approaches to study the importance of resilience factors is to examine their predictive value for resilience and, less common, recovery trajectories using a GMM approach, with some studies employing classical multinomial logistic regression analyses and others adopting a three-step approach accounting for uncertainty in class assignments[38]. Some of those studies also account for the well-established predictive value of other variables such as pre-stressor mental health[39,40] and previous stressor exposure[41]. To note, we use the terms ‚predict' and ‚predictor' to refer to independent variables in regression models as this is often done in Psychology[42]. We are not referring to proper prediction modeling and causal relationships when we use this term.

For a long time, studies on resilience factors focused on individual resilience factors such as dispositional variables (e.g., optimism[15], hardiness[43]) or beliefs (e.g., self-efficacy[16], control beliefs[44]). Consequently, previous qualitative reviews on resilience factors[17,19,45,46] comprised a large number of potentially resilience-promoting traits and beliefs. However, a major short-coming of research into individual resilience factors is the missing conceptual clarity[17,47,48] with substantial empirical and conceptual overlaps of different factors (e.g., self-efficacy and locus of control, meaning in life and spirituality)[49]. As many individual resilience factors are examined in single studies without studying their incremental validity above other (resilience) factors, knowledge on their unique value to predict resilient responses is rare, with some findings suggesting a decreasing relevance in joint models[25,27,49,50]. This complicates basic research into resilience[49,51], the development and evaluation of resilience interventions as well as the monitoring of resilience factors as part of public mental health surveillance[7].

Research into social resilience factors mostly focused on perceived and, less often, received, enacted or structural social support[18,52], with perceived

social support referring to the perceived availability and adequacy of social support, while received social support focuses on the social support actually provided by others[53]. Enacted social support refers to actions an individual takes to help others[18]. Structural social support reflects the size and strength of one's support network. Most meta-analyses find perceived social support to share the strongest link with stress-related mental symptoms[18,54], however, the longitudinal association of perceived social support and mental health has recently been challenged by a meta-analysis suggesting that longitudinal associations may result from statistical artifacts due to inappropriate modeling of longitudinal data[55]. Other social resilience factors like family cohesion, social connectedness, or social participation have rarely been examined in resilience research.

Societal resilience factors have been studied even more rarely, with most research having been conducted in the fields of public health[56,57] and security research[58]. A recent review suggested to differentiate between contextual and target factors[59], with contextual factors being those fundamental aspects that characterize communities and societies (e.g., macroeconomic characteristics, income [in]equality, pollution, access to natural spaces, trust in institutions) and target factors being those aspects that could be (more easily) addressed by political measures and interventions (e.g., perceived safety, availability of infrastructure, job security)[56]. Other classifications[14] differentiate factors related to the built or natural environment, while all social processes fall into the category of social resilience factors. So far, a consensus on the categorization of societal resilience factors is missing and they have rarely been included in studies on individual responses to major stressors.

The current review aims at providing a systematic overview on multilevel resilience factors in the face of societal challenges in member states of the Organization for Economic Cooperation and Development (OECD). We examine OECD member states as these countries are mostly high-income countries that describe themselves as democracies[60], which allows for a minimum of comparability between studies. In contrast to previous research[17,57,61–65], the current review will examine multiple resilience factors at the individual, social and societal level across different types of societal challenges. In line with previous reviews[17,33,66], we focus on studies employing different types of trajectory modeling[35,36]. This decision was based on the fact that these models represent the most common approaches to study resilience as an outcome[32,67], and allow to examine the longitudinal association of previously assessed resilience factors with post-stressor changes in mental health, while limiting the amount of between-study heterogeneity. In contrast to other statistical approaches, trajectory modeling allows for contrasting resilient with non-resilient responses, which enables the identification of predictors of resilient responses. Moreover, analyses and conclusions on the importance of resilience factors will be based on longitudinal studies, partly also including pre-stressor data, and assess whether there is evidence for the incremental validity of resilience factors beyond sociodemographic variables and other resilience factors.

Thereby, we aim at answering the following research questions: (1) Which societal-level challenges and crises have been examined in OECD member states? (2) What kind of mental health outcomes have been examined to study consequences of those challenges? (3) What kind of individual, social and societal resilience factors and mechanisms have been examined in face of those societal challenges? Is there a trend towards specific resilience factors and mechanisms being examined more often in the context of specific stressors? (4) What is the evidence level for each resilience factor and mechanism, and what can we conclude on the incremental validity of each factor beyond sociodemographic variables and other resilience factors? (5) What are study, participant and contextual factors impacting on the evidence ratings for resilience factors? Additionally, we identify key knowledge gaps when studying resilience factors and mechanisms in the context of societal challenges.

## Methods
This systematic review is reported in line with the Preferred Reporting Items for Systematic Reviews and Meta-Analyses (PRISMA)[68]. Differences

between the prospective preregistration of the review on June 22, 2023 (preregistration-ID: https://doi.org/10.17605/OSF.IO/GWJVA) and the final review are presented in Supplementary Note 2. Most importantly, the project developed from a scoping review to a systematic review and the rating scheme for resilience factors was amended during the review and revision process.

### Search strategy
The search strategy for this review builds on a larger review project (pre-registration-ID: https://doi.org/10.17605/OSF.IO/A9HWN; results will be reported elsewhere). Five databases were searched from 2004 to present (last update: August 2, 2023), including APA PsycNet (incl. PsycInfo, PsycArticles, PsycExtra), Embase (incl. PubMed and EmbaseCore), PTSDPubs, Scopus, and the Web of Science Core Collection. The primary search contained three clusters with search terms related to (i) stress exposure (e.g., trauma, stress, life event), (ii) mental health (e.g., anxiety, mental distress, wellbeing), and (iii) trajectory modeling (e.g., latent growth, trajectory). Terms within one cluster were linked using the Boolean operator *OR* and clusters were combined using the operator *AND* (see Supplementary Note 3 for the full search strategy). Moreover, reference lists of related systematic reviews[17,33] and included primary studies were checked for eligible studies.

### Search criteria
Eligible studies were longitudinal observational studies examining adult individuals (≥18 years) from the civil general population, not recruited from military or clinical contexts, who were exposed to all kinds of societal challenges and crises in member countries of the OECD. In line with recent studies in the field of public health[2], such stressors include pandemics, wars and armed conflicts, the climate crisis, and natural disasters (see Supplementary Note 4 for a full list of potentially eligible societal challenges). Studies needed to examine trajectories of mental health by means of GMM[35] (or comparable methodological approaches to trajectory modeling aiming at identifying different patterns of mental health over time) and investigate individual, social or societal resilience factors as their predictors (i.e., as an independent variable in a regression analysis). All methods to examine predictor variables were eligible (e.g., three-step and standard multinomial regression analyses[38]). The classification of resilience factors was based on previous reviews[17,19,22,45,46] in the field and limited to multilevel psychosocial resources. Notably, other pre-stressor factors well known to predict post-stressor mental health (e.g., pre-stressor mental health[39,40], lifetime stressor exposure[41]) were not examined as resilience factors. However, as there is no finite list of resilience factors, we also included all kinds of factors that were discussed as potentially health- or resilience-promoting by primary study authors. Moreover, some factors (e.g., education, income, family status or socioeconomic status) could either be classified as sociodemographic characteristic or resilience factor. In these cases, variables were included as resilience factors when they were either potentially modifiable by individual or systemic interventions[13] (e.g., education, income) or might provide a proxy measure of rather well-established resilience factors (e.g., family status or living with a partner as indicators of available social support). As a consensus definition of the differentiation between resilience factors and mechanisms is still missing and a matter of ongoing debates[22,23], this distinction was based on two criteria: First, we reviewed the labeling provided by primary study authors; second, we examined whether factors were examined that are frequently discussed as potential resilience mechanisms based on landmark reviews in the field[11,14,22,23,69,70]. Studies needed to include ≥ 300 participants and to comprise at least three assessment waves, with no requirement for pre-stressor data. However, stressor exposure and the first assessment wave needed to be at most four years apart.

### Study selection
After de-duplication using Zotero[71], titles, abstracts, and full texts were assessed independently by two reviewers in Rayyan[72]. Interrater reliability was substantial at title/abstract level (*kappa* = 0.68) and full text level

(*kappa* = 0.75). At both screening stages, disagreements were resolved through discussion or consultation of a senior team member.

## Data extraction

We developed a customized data extraction sheet for this review (available from OSF: https://osf.io/9xwyu/). All data of eligible primary studies were extracted by one reviewer and checked by a second, with disagreements being resolved through discussion or consultation of senior team members. Data extraction focused on sample characteristics, types of societal challenges, and trajectories identified using trajectory modeling, and included information needed for later evidence ratings for resilience factors. Moreover, we extracted information needed for later quality appraisal (i.e., representativeness, outcome assessment, statistical model). Data were extracted for the broader outcome categories of mental distress (i.e., general distress, depressive symptoms, anxiety symptoms, posttraumatic stress symptoms, stress symptoms) and positive mental health (i.e., life satisfaction, personal growth, mental health related quality of life, well-being). Resilience factors were classified as either representing individual, social or societal resources by one reviewer, with individual resources being psychological dispositions, beliefs, or capabilities. Social factors were resources that were perceived or available in one's nearer social environment (e.g., family, friends), while societal factors were resources in the wider environment or the whole society (e.g., trust in authorities, legal protection; see Supplementary Note 5 for details on this classification). The decision on resilience factor level was checked by a second reviewer, with all disagreements being discussed and solved in the review team.

## Quality appraisal

Study quality was assessed by two team members as an indicator of risk of bias using a modified version of the Newcastle-Ottawa Scale (NOS[73,74]), assessing bias from (1) selection, (2) comparability, (3) outcome assessment, (4) reporting of methodological details, and (5) quality of trajectory modeling (i.e., constraints of variances across classes and slopes within one class; see OSF project: https://osf.io/9xwyu/). Based on the number of items that could be assessed per study, we calculated an overall study quality rating ranging from 0% to 100%.

## Data synthesis

After descriptive synthesis, data were analyzed in a qualitative manner extending a rating scheme that was developed for two previous reviews on resilience factors[17,75] and making use of the rationale of effect size-informed vote counting[76] for evidence synthesis. All analyses were performed in IBM SPSS statistics version 29[77].

For this systematic review, we aimed at providing a qualitative synthesis of findings amended by insights from additional quantitative analyses. Although we had estimates of odds ratios (ORs) or comparable regression coefficients from most of the included primary studies, these coefficients were controlled for highly heterogeneous variables (i.e., the number and nature of control variables varied substantially between studies). This prevented the use of standard meta-analysis for synthesis[78,79]. Thus, all quantitative analyses are based on non-parametric statistical tests that can only be viewed as an add-on to our qualitative summary and need further replication using standard meta-analyses based on more homogeneous primary studies.

Table 1 presents the rating scheme that was used for each (individual, social and societal) resilience factor and each mental health outcome (i.e., in cases where one resilience factor was examined as a predictor of three mental health outcomes, three ratings were performed). We differentiate between three levels of evidence: a resilience factor shows a significant association with a favorable or unfavorable trajectory, (1) without control of any other variable ( + *or* -); (2) under control of sociodemographic variables (++ *or* --); or (3) under control of sociodemographic variables *and* at least one other resilience factor (+++ *or* ---). Moreover, when we found no evidence for a link between resilience factors and favorable or unfavorable trajectories, we differentiated whether these null effects were found without any control (o),

controlled for sociodemographic variables (oo), or controlled for sociodemographic variables *and* other resilience factors (ooo). Based on the criticism of statistical significance tests[80], this approach was amended by information derived from effect sizes of ORs or regression coefficients. Thereby, we acknowledge the fact that especially large samples are at risk for overidentification of resilience factors that only have a small association with resilient responses. Using transformations for dichotomized outcomes in meta-analyses[81], we specified at each level of our rating scheme whether effects were very small (A: *OR* < 1.44 or *OR* > 0.70), small (B: *OR* 1.44–2.47 or *OR* 0.41–0.70), medium (C: *OR* 2.48–4.26 or *OR* 0.24–0.40), or large (D: *OR* ≥ 4.27 or *OR* ≤ 0.23) according to Cohen[82]. A detailed rating guideline including the handling of special cases was uploaded to OSF.

To note, for some (resilience) factors it is not yet clear as to whether they protect individuals from potentially harmful effects of stressors, or may also make them more sensitive to stress, with primary studies showing contradicting findings even for the same stressor[83,84]. The concept of regulatory flexibility assumes that the match between individual resources (e.g., cognitive and emotional coping strategies) and contextual demands is essential for factors being either adaptive or maladaptive, with mismatches accounting for between-study differences[24,25]. Thus, our rating scheme also included information on potential resilience factors showing associations with less favorable responses to stress (levels - to ---). Similarly, also at this side of the rating scale, effect estimates were rated for their effect size.

For our primary analyses, we focused on the comparison of resilient trajectories versus unfavorable responses (i.e., delayed, chronic or other clearly less favorable responses; see Fig. 1), while our secondary analyses compared recovery (or emergent resilience) trajectories with less favorable responses (i.e., delayed, chronic or other clearly less favorable patterns).

In the next step, we aimed at examining whether evidence ratings for resilience factors were associated with study and participant characteristics (e.g., study design, sample mean age, gender [im]balance, sample representativeness, number of assessment waves, number of variables and resilience factors included for modeling) as well as contextual factors (e.g., stressor type). For resilience factors with multiple ratings being available from a single study, a median was calculated for analyses to reflect the average rating for the respective factor. We used Fisher-Freeman-Halton exact tests as equivalent of $\chi^2$ tests with small counts per cell[85], Kruskal–Wallis tests as non-parametric equivalent of analysis of variances[86], Mann–Whitney U tests to compare ratings between independent samples[87], and Spearman's rank correlations to examine the link between participant characteristics and evidence ratings[88]. For overall significant and close-to-significant Freeman-Halton tests, we descriptively report on differences between counts and expected counts with a focus on the most prominent deviations. For those descriptions, we did not employ post-hoc tests to limit the number of statistical tests in our review. In cases where Freeman-Halton tests provided non-significant results ($p > .10$), we do not provide a summary of numerical differences to not overemphasize differences that likely result from chance. For Fisher-Freeman-Halton and Kruskal–Wallis tests we used a Monte-Carlo approach with 10,000 samples. For all statistical tests, we report 95% confidence intervals (CIs) of test statistics.

In our sensitivity analyses, we examined the importance of study quality and timing. For this purpose, we used Spearman's rank correlations and examined the association between study quality ratings and evidence ratings with a significant correlation coefficient suggesting a relevant impact of study quality. Using the same approach, we examined whether there was a link between evidence ratings and (i) the time interval between the last pre-stressor assessment and occurrence of the stressor; (ii) the time interval between stressor exposure and the first post-stressor assessment; and (iii) the time interval between stressor exposure and last assessment.

## Reporting summary

Further information on research design is available in the Nature Portfolio Reporting Summary linked to this article.

**Table 1 | Rating of evidence levels of resilience factors**

| Category | OR | Level of evidence (assessed per resilience factor and outcome) |
|---|---|---|
| **+++** | D) OR ≥ 4.27<br>C) OR 2.48 – 4.26<br>B) OR 1.44 – 2.47<br>A) OR < 1.44 | The resilience factor is significantly associated with resilient outcomes (i.e., resilience trajectories vs. delayed, chronic or other clearly less favorable responses) under control of **other resilience factors _and_ sociodemographic variables**. |
| **++** | D) OR ≥ 4.27<br>C) OR 2.48 – 4.26<br>B) OR 1.44 – 2.47<br>A) OR < 1.44 | The resilience factor is significantly associated with resilient outcomes (i.e., resilience trajectories vs. delayed, chronic or other clearly less favorable responses) under control of **sociodemographic variables**. |
| **+** | D) OR ≥ 4.27<br>C) OR 2.48 – 4.26<br>B) OR 1.44 – 2.47<br>A) OR < 1.44 | The respective resilience factor is significantly associated with resilient outcomes (i.e., resilience trajectories vs. delayed, chronic or other clearly less favorable responses) **without control** of other variables. |
| **ooo** | D) + OR ≥ 4.27<br>C) + OR 2.48 – 4.26<br>B) + OR 1.44 – 2.47<br>A) + OR < 1.44 | No significant association of the respective resilience factor with any outcome (i.e., resilience trajectories, less favorable trajectories) under control of **other resilience factors _and_ sociodemographic variables**. |
| **oo** | D) + OR ≥ 4.27<br>C) + OR 2.48 – 4.26<br>B) + OR 1.44 – 2.47<br>A) + OR < 1.44 | No significant association of the respective resilience factor with any outcome (i.e., resilience trajectories, less favorable trajectories) under control of **sociodemographic variables**. |
| **o** | D) + OR ≥ 4.27<br>C) + OR 2.48 – 4.26<br>B) + OR 1.44 – 2.47<br>A) + OR < 1.44 | No significant association of the respective resilience factor with any outcome (i.e., resilience trajectories, less favorable trajectories) **without control** of other variables. |
| **o** | mixed (+ and -) | No significant association of the respective resilience factor with any outcome (i.e., resilience trajectories, less favorable trajectories) **without control** of other variables. |
| **oo** | mixed (+ and -) | No significant association of the respective resilience factor with any outcome (i.e., resilience trajectories, less favorable trajectories) under control of **sociodemographic variables**. |
| **ooo** | mixed (+ and -) | No significant association of the respective resilience factor with any outcome (i.e., resilience trajectories, less favorable trajectories) under control of **other resilience factors _and_ sociodemographic variables**. |
| **o** | A) - OR > 0.70<br>B) - OR 0.41 – 0.70<br>C) - OR 0.24 – 0.40<br>D) - OR ≤ 0.23 | No significant association of the respective resilience factor with any outcome (i.e., resilience trajectories, less favorable trajectories) **without control** of other variables. |
| **oo** | A) - OR > 0.70<br>B) - OR 0.41 – 0.70<br>C) - OR 0.24 – 0.40<br>D) - OR ≤ 0.23 | No significant association of the respective resilience factor with any outcome (i.e., resilience trajectories, less favorable trajectories) under control of **sociodemographic variables**. |
| **ooo** | A) - OR > 0.70<br>B) - OR 0.41 – 0.70<br>C) - OR 0.24 – 0.40<br>D) - OR ≤ 0.23 | No significant association of the respective resilience factor with any outcome (i.e., resilience trajectories, less favorable trajectories) under control of **other resilience factors _and_ sociodemographic variables**. |
| **-** | A) OR > 0.70<br>B) OR 0.41 – 0.70<br>C) OR 0.24 – 0.40<br>D) OR ≤ 0.23 | The respective resilience factor is significantly associated with less favorable outcomes (i.e., delayed, chronic or other clearly unfavorable responses vs. resilience trajectories) **without control** of other variables. |
| **--** | A) OR > 0.70<br>B) OR 0.41 – 0.70<br>C) OR 0.24 – 0.40<br>D) OR ≤ 0.23 | The respective resilience factor is significantly associated with less favorable outcomes (i.e., delayed, chronic or other clearly unfavorable responses vs. resilience trajectories) under control of **sociodemographic variables**. |
| **---** | A) OR > 0.70<br>B) OR 0.41 – 0.70<br>C) OR 0.24 – 0.40<br>D) OR ≤ 0.23 | The respective resilience factor is significantly associated with less favorable outcomes (i.e., delayed, chronic or other clearly unfavorable responses vs. resilience trajectories) under control of **other resilience factors _and_ sociodemographic variables**. |

Evidence levels ranging from +++ (= most favorable level of evidence for the respective factor from a single primary study, i.e., the respective factor showed incremental validity beyond other resilience factors and sociodemographic variables with the highest level of control for other variables) to --- (= least favorable level of evidence for the respective factor from a single primary study, i.e., there is evidence for the respective factor being associated with unfavorable trajectories with the highest level of control for other variables). Rating categories o, oo, and ooo represent statistically non-significant findings with different levels of control. Moreover, at each level, we further coded effect sizes reported for the respective resilience factor based on regression coefficients, which could either be very small (A: $OR < 1.44$ or $OR > 0.70$), small (B: $OR$ 1.44–2.47 or $OR$ 0.41–0.70), medium (C: $OR$ 2.48–4.26 or $OR$ 0.24–0.40), or large (D: $OR ≥ 4.27$ or $OR ≤ 0.23$). In the case of null effects, we further differentiated whether the effect sizes indicated a consistent trend (+ = consistently positive; - = consistently negative) or were mixed (+ and - = positive and negative). Due to the overall debate of the use of ORs in the context of non-dichotomous outcomes[132], we favored evidence from statistical tests over effect sizes, i.e., ratings of effect sizes were nested within categories assessing statistical significance.

In cases where coding in primary studies was inverse, that is, higher scores indicated lower levels of the respective resilience factor (e.g., poor social support), the respective coding was inverted for our rating scheme in a way that higher scores indicated higher levels of the respective resilience factor.

OR odds ratio, + effect estimates with positive sign, – effect estimates with negative sign.

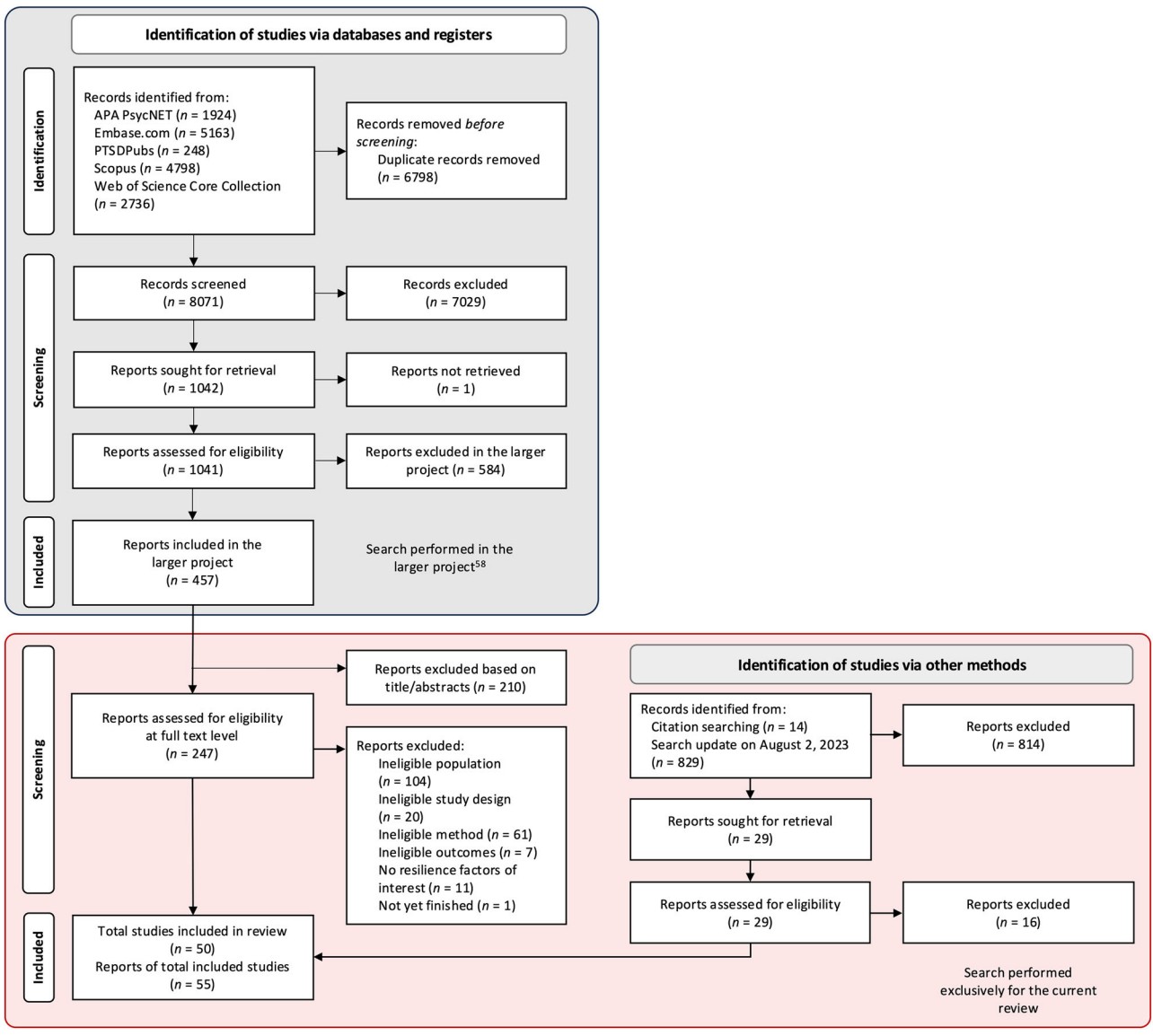

**Fig. 2 | PRISMA flowchart.** Flowchart according to the Preferred Reporting Items for Systematic Reviews and Meta-Analyses (PRISMA)[68]. The gray part of the figure reflects the search performed for the larger project (preregistration-ID: https://doi.org/10.17605/OSF.IO/A9HWN), the red part represents the search for this review.

## Results

### Search outcomes

For the larger project, databases yielded 14,869 records with 6798 being removed as duplicates (see Fig. 2). Of 8071 records screened at title and abstract level, 1041 were assessed at full-text level. Resulting in 457 eligible records for the larger project that were transferred to our project for potential inclusion. Of those, 210 were excluded based on titles and abstracts as stressors were ineligible, the remaining 247 records were assessed at full-text level with 42 being eligible. Another 14 records were obtained via citation searching and 829 records were identified in our search update in August 2023, of which 29 were assessed at full-text level, with 13 being eligible for inclusion. Taken together, this resulted in 50 eligible primary studies (from 55 records).

### Study and sample characteristics

Table 2 presents the characteristics of the included studies published between 2009 and 2023. The studies were performed in 15 solely high-income OECD countries, including USA (18 studies), the United Kingdom (9 studies), and Australia (4 studies).

Samples sizes of primary studies ranged between 360 and 65,818 participants. Thirty-seven studies examined adults from the general

population, while specific high-risk populations (e.g., healthcare professionals, police staff, migrants, low-income mothers) were examined in 9 studies. Another four studies examined selective subsamples from the general population without particular risk (e.g., university staff, tourists). Only a small share (11 studies) was representative of the respective target population with most studies using convenience samples. Mean sample age ranged between 20.01 and 78.69 years (weighted mean: 48.58 years), with 13.4% to 100% (weighted mean: 53.05%) of the respondents self-identifying as women. Attrition was insufficiently reported in many studies, but attrition rates were high for most studies (i.e., up to 99%[89]), indicating decreasing data quality over time.

The vast majority of studies (84.0%) used variants of growth mixture modeling (GMM), which varied with respect to their restrictiveness. Twenty-five studies (50.0%) employed GMM allowing for within class variations of intercepts and slopes, while 17 studies (34.0%) used variants of latent class growth modeling as a more restrictive approach fixing intercepts and/or slopes within classes (see Table 2 for all models). Only seven studies (14.0%) allowed free slopes within trajectories, while no study allowed for different variances between trajectories. The most common approach to examine the predictive value of resilience factors were different variants of multinomial logistic regression models, which were used in all but one

## Table 2 | Characteristics of included primary studies

| Study | Country | Societal challenge | N modeling | Population (mean age, % female) | No. of assessments (first and last timepoint) | Trajectories modeling approach | Outcomes with no. of trajectories per outcome | Resilience factors examined | Statistical model used to examine resilience factors | Overall quality assessment (NOS) |
|---|---|---|---|---|---|---|---|---|---|---|
| Allinson 2023[133,a] | Australia | Movements of refugees (immigration to Australia) | 1496 | Humanitarian migrants: 38.55 ± 12.97, 29.6% | 5 (2013–2018) (−) | Latent class growth analysis (LCGA) | General distress (3), PTSD symptoms (4) | Education, financial stress | Multinomial logistic regression model | 66.67% |
| Batterham 2021[134] | Australia | Pandemics (COVID-19 pandemic) | 1296 | General population: 46.0 ± 17.30, 50.1% | 7 (03/2020 – 06/2020) (−) | Growth mixture models (GMM) | Anxiety symptoms (4), depressive symptoms (3) | Education, having a partner, living with family/others | Multinomial logistic regression model | 72.22% |
| Bayes-Marin 2023[90] | Spain | Pandemics (COVID-19 pandemic) | 5530 | Middle-aged adults from the general population: 51.17 ± 6.93, 67.4% | 9 (11/2018 and 12/2019 – 02/2022) (+) | Growth mixture models (GMM) | General distress (3), positive mental health (3) | Education, overall (adaptive/functional) coping, household/family income, living with family/others, rural region | Multinomial logistic regression model | 55.56% |
| Carr 2022[135] | UK | Pandemics (COVID-19 pandemic) | 2241 | University staff and postgraduate students from the general population: NR, 70.6% | 27 (04/2020 – 04/2021) (−) | Growth mixture models (GMM) | Anxiety symptoms (4), depressive symptoms (4) | Having a partner, living with family/others | R3STEP procedure in Mplus | 77.78% |
| Ellwardt 2021[91,c] | UK | Pandemics (COVID-19 pandemic) | 15914 | General population: NR, NR | 9 (pre-Covid – 05/2021) (+) | Latent class mixture modeling (LCMM) | General distress (4) | Having a partner | Multinomial logistic regression model | 55.56% |
| Feder 2016[136,b] | USA | Terrorist attacks (9/11 terrorist attacks) | 1874 police responders; 2613 non-traditional responders | Police responders: 41.7 ± 6.9, 14.6%; non-traditional responders: 46.2 ± 9.6, 13.7% | 4 (median of T1: 2.8 years after 9/11; until T4: 12.2 years after 9/11) (−) | Growth mixture modeling (GMM) | PTSD symptoms (4/5) | Active coping, education, income, positive emotion-focused coping, purpose in life, religious coping, social coping, having a partner, perceived social support, structural family support, structural work support | Multinomial logistic regression model | 44.44% |
| Fogden 2020[137,a] | Australia | Movements of refugees (immigration to Australia) | 1495 | Humanitarian migrants: 36.86 ± 12.66, 29.6% | 4 (10/2013 – 02/2017) (−) | Latent class growth analyses (LCGA) | General distress (4), PTSD symptoms (4) | Living with family/others | Multinomial logistic regression model | 66.67% |
| Galovski 2018[138] | USA | Systemic stressor (violence during civil unrest) | 558 | General population & law enforcement officers: 40.66 ± 13.08, 41.6% | 3 (11/2014 – 2/2016) (−) | Latent class growth modeling (LCGM) | Depressive symptoms (4), PTSD symptoms (4) | Education, income, perceived social support | R3STEP procedure in Mplus | 66.67% |
| Gambin 2023[139] | Poland | Pandemics (COVID-19 pandemic) | 1100 | General population: 44.70 ± 15.82 | 5 (5/2020 – 4/2021) (−) | Growth mixture modeling (GMM) | General distress (4) | Education, empathic concern, financial situation, overall emotion regulation, perspective taking, having a partner, living with family/others, received social support | R3STEP procedure in Mplus | 83.33% |
| Goodwin 2020[140] | Japan | Environmental or natural disasters (Great East Japan Earthquake + Level 7 nuclear accident) | 2599 | General population: 54.63 ± 15.92, 53.9% | 5 (2012 – 2016) (−) | Growth mixture modeling (GMM) | General distress (4) | Perceived social support | Multinomial logistic regression model | 66.67% |
| Gruebner 2016[141,d] | USA | Environmental or natural disasters (Hurricane Ike) | 561 | General population: NR, 58.8% | 3 (11/2008 – 01/2010) (−) | Latent class growth analysis (LCGA) | Depressive symptoms (4), PTSD symptoms (4) | Education, received social support, perceived collective efficacy | Multivariable logistic regression model | 66.67% |
| Hemi 2023[26] | Israel | Pandemics (COVID-19 pandemic) | 571 | General population: 43.30 ± 14.48, 79% | 3 (4/2020 – 6/2021) (−) | Growth mixture modeling (GMM) | Anxiety symptoms (4), depressive symptoms (2) | Education, cognitive flexibility, coping flexibility, perceived social support | Multinomial logistic regression model | 55.56% |

## Table 2 (continued) | Characteristics of included primary studies

| Huber 2022[142] | USA | Terrorist attacks (9/11 terrorist attacks) | 37545 | General population: NR, 53.6% | 4 (9/2003 – 1/2016) (–) | Growth mixture modeling (GMM) | PTSD symptoms (4) | Education, having a partner, household/family income | R3STEP procedure in Mplus | 77.78% |
|---|---|---|---|---|---|---|---|---|---|---|
| Hyland 2021[83] | Ireland | Pandemics (COVID-19 pandemic) | 1041 | General population: NR, 51.5% | 4 (3/2020 – 12/2020) (–) | Growth mixture modeling (GMM) | General distress (4) | Empathy, income, internal locus of control, tolerance of uncertainty, living with family/others, rural region | Multinomial logistic regression model | 72.22% |
| Iob 2020[89] | UK | Pandemics (COVID-19 pandemic) | 51417 | General population: 48.80 ± 16.80, 51.1% | 7 (03/2020 – 05/2020) (–) | Growth mixture modeling (GMM) | Depressive symptoms (3) | Perceived social support, socioeconomic status | Multivariate logistic regression model | 66.67% |
| Johannesson 2015[143] | Sweden | Environmental or natural disasters (2004 Indian Ocean tsunami) | 3518 | Swedish tourists: 49.50 ± 14.00, 59.0% | 3 (T1: 14 months post disaster – T3: six years post-disaster) (–) | Growth mixture model (GMM) | PTSD symptoms (4) | Education, having a partner, received social support | Multinomial logistic regression model | 55.56% |
| Jordan 2023[144] | UK | Pandemics (COVID-19 pandemic) | 585 | Health and social care staff: 43.56 ± 10.54, 83.0% | 4 (11/ 2020 – 08/2021) (–) | Growth mixture models (GMM) | Anxiety symptoms (2), depressive symptoms (2), PTSD symptoms (2) | Structural work support | R3STEP procedure in Mplus | 72.22% |
| Joshi 2021[145] | Canada | Pandemics (COVID-19 lockdown) | 579 | University faculty and staff: NR, 79.42% | 6 (04/2020 – 11/2020) (–) | Growth mixture model (GMM) | Depressive symptoms (2) | Education, emotion-focused coping, overall (adaptive/functional) coping, problem-focused coping, having a partner | Multivariable analysis | 55.56% |
| Kimhi 2021[146] | Israel | Pandemics (COVID-19 pandemic) | 804 | General population: 44.65 ± 15.40, 48.0% | 3 (05/2020 – 10/2020) (–) | Growth mixture modeling (GMM) | Anxiety symptoms (4), depressive symptoms (4) | Household/family income | Multinomial logistic regression model | 72.22% |
| Ko 2021[147] | USA | Terrorist attacks (9/11 terrorist attacks) | 30839 | General population: NR, 38.8% | 4 (2003 –2016) (–) | PROC TRAJ SAS (group-based modeling of longitudinal data) | General distress (5) | Education, income | Logistic regression model | 55.56% |
| Li 2023[92] | Australia | Environmental or natural disasters (climate-related disaster like floods, bushfires, or cyclones) | 1357 | General population: 45.11 ± 17.86, 52.0% | 8 (2010 –2019) (+) | Stata traj (group-based multi-trajectory modeling) | Positive mental health (3) | Education, household/family income, living with family/others, social participation, community attachment, local house value, milder temperature, neighborhood environment, rural region, warmer temperature | Multinominal logistic regression model | 50.00% |
| Lopez-Castro 2022[148] | USA | Pandemics (COVID-19 pandemic) | 1206 | General population: 39.36 ± 14.09, 52.1% | 4 (04/2020 – 07/2021) (–) | Growth mixture modeling (GMM) | PTSD symptoms (4) | Education, income | Bias-adjusted three-step approach in R | 83.33% |
| Lowe 2013[93] | USA | Environmental or natural disasters (Hurricane Katrina + Hurricane Rita) | 386 | Low-income mothers: 26.40 ± 4.43, 100% | 3 (08/2005 – 03/2010) (+) | Latent class growth analyses (LCGA) | General distress (6) | Perceived social support | One-way ANOVA + Bonferroni-corrected post hoc tests/ χ² test | 44.44% |
| Lowe 2015[149,150,d] | USA | Environmental or natural disasters (Hurricane Ike) | 658 | General population: NR, 59.9% | 3 (11/2008 – 4/2010) (–) | PROC TRAJ SAS (group-based mixture modeling procedure) / Latent class growth analysis (LCGA) | General distress (NA) | Education, received social support, perceived collective efficacy | Hierarchical logistic regression model | 66.67% |
| Lowe 2020[94] | USA | Environmental or natural disasters | 885 | Low-income mothers: 25.19 ± 4.45, 100% | 4 (11/2003 – 12/2018) (+) | Latent class growth analyses (LCGA) | PTSD symptoms (3) | Having a partner, perceived social support | R3STEP procedure in Mplus | 66.67% |

## Table 2 (continued) | Characteristics of included primary studies

| | | | | | | | | | | |
|---|---|---|---|---|---|---|---|---|---|---|
| | | (Hurricane Katrina) | | | | | | | | |
| Lu 2022[95] | France | Pandemics (COVID-19 pandemic) | 613 | General population: 58.10 ± 12.10, 50.4% | 3 (11/2014 – 10/2020) (+) | Latent Class Mixed Models (LCMM) | Anxiety symptoms (2), depressive symptoms (3) | Education, having a partner, rural region | Multinomial logistic regression model | 66.67% |
| Mandavia 2019[96] | USA | Financial or economic crises (Great recession) | 1172 | Older-aged adults from the general population: 60.67± 6.86, 64.6% | 3 (2006 – 2014) (+) | Growth mixture modeling (GMM) | Depressive symptoms (3) | Education, perceived social support | Logistic regression model | 33.33% |
| Matovic 2023[151] | Canada | Pandemics (COVID-19 pandemic) | 645 | Older-aged adults from the general population: 78.69 ± 5.67, 73.1% | 4 (05/2020 – 05/2021) (−) | Group-based trajectory modeling (GBTM) | General distress (3) | Education, financial poverty, living with family/ others | Multinomial logistic regression model | 66.67% |
| McPherson 2021[152] | UK | Pandemics (COVID-19 lockdown) | 1946 | Community-dwelling older adults: NR, 70.0% | 4 (23/3/2020 – 25/6/2020) (−) | Growth mixture modeling (GMM) | Anxiety symptoms (4), depressive symptoms (4), PTSD symptoms (4) | Meaning in life, having a partner, living with family/others, perceived social support, rural region | Multinomial logistic regression model | 61.11% |
| Meli 2022[153] | USA | Pandemics (COVID-19 pandemic) | 404 | General population: 39.27 ± 13.25, 45.40% | 3 (04/2020 – 04/2021) (−) | Growth mixture modeling (GMM) | Anxiety symptoms (4), depressive symptoms (2) | Education | Least squares regression analyses within the LGMM framework | 61.11% |
| Moulin 2023[154] | France | Pandemics (COVID-19 pandemic) | 681 | General population: 46.6 ± 15.34, 78.56% | 4 (05/2020 – 04/2021) (−) | Cluster analysis using kml3d | General distress (2) | Religious practice, perceived social support | Logistic regression model | 61.11% |
| Nandi 2009[155] | USA | Terrorist attacks (9/11 terrorist attacks) | 2282 | General population: 44.7, 54.81% | 4 (03/2002 – 11/2005) (−) | Semi-parametric group-based modeling (type of latent growth curve analysis) | Depressive symptoms (5) | Education, having a partner, household/family income, perceived social support | Adjusted trajectory models from semi-parametric group-based modeling | 61.11% |
| Oe 2016[156] | Japan | Environmental or natural disasters (Fukushima Daiichi Nuclear Power Plant accident) | 12371 | General population: NR, 57.2% | 3 (01/2012 – 02/2014) (−) | Group-based trajectory modeling (GBTM) using PROC TRAJ SAS | General distress (4) | Structural social support (general) | Logistic regression model | 55.56% |
| Orcutt 2014[97] | USA | Non-environmental disasters (Campus Mass Shooting) | 660 | College women: 20.01 ± 2.56, 100% | 7 (08/2006 – 02/2008) (+) | Growth mixture modeling (GMM) | PTSD symptoms (4) | Emotional clarity, overall emotion regulation | Multinomial logistic regression model | 66.67% |
| Pellerin 2022[50] | France | Pandemics (COVID-19 pandemic) | 1399 | General population: 43.4 ± 12.00, 87.8% | 5 (04/2020 – 05/2020) (−) | Growth mixture modeling (GMM) | Anxiety symptoms (3), depressive symptoms (4) | Gratitude, hope, optimism, peaceful disengagement, psychological flexibility, self-efficacy, wisdom, living with family/others, relationship quality, environment quality | Multinomial logistic regression model | 66.67% |
| Pierce 2021[98,c] | UK | Pandemics (COVID-19 pandemic) | 18321 | General population: NR, 51.6% | 6 (2018 – 10/2020) (+) | Latent Class Mixed Models (LCMM) | General distress (5) | Having a partner, neighborhood environment | Three-step procedure (including multinominal logistic regression) | 94.44% |
| Pietrzak 2014[,123] | USA | Terrorist attacks (9/11 terrorist attacks) | 4035 police responders; 6800 non-traditional responders | Police responders: 41.2 ± 6.6, 14.67%; non-traditional responders: 45.3 ± 9.6, 13.43% | 3 (average of T1: 3.3 years after 9/11 – T3: 5.3 years after 9/11) (−) | Growth mixture modeling (GMM) | PTSD symptoms (4/6) | Education, income, having a partner, structural family support, structural work support | Multinomial logistic regression model | 44.44% |
| Piscitello 2020[157] | USA | Environmental or natural disasters (Hurricane Katrina) | 360 | Mothers from general population: 38.9 ± 7.6, 100.00% | 4 (T1: 3-7 months post Hurricane Katrina – T4: 25-27 months post-Hurricane Katrina) (−) | Latent class growth analysis (LCGA) | PTSD symptoms (3) | Overall (adaptive/functional) coping, perceived social support | Multinomial logistic regression model | 44.44% |

## Table 2 (continued) | Characteristics of included primary studies

| Probst-Hensch 2023[158] | Switzerland | Pandemics (COVID-19 pandemic) | 6396 | General population: 57.66 ± 14.17, 55.68% | 19 (07/2020 – 12/2021) (−) | Group-based trajectory modeling (GBTM) | Depressive symptoms (3) | Education, household/family income, living with family/others | Multinomial logistic regression model | 55.56% |
|---|---|---|---|---|---|---|---|---|---|---|
| Pruncho 2021[99] | USA | Financial or economic crises (the Great Recession) | 3566 | Older-aged adults from the general population: 60.79 ± 7.10, 63.7% | 5 (2006 –2019) (+) | Latent class growth model (LCGM) using PROC TRAJ SAS | Depressive symptoms (4) | Income | Multinomial logistic regression model | 55.56% |
| Qi 2022[100] | Netherlands | Pandemics (COVID-19 pandemic) | 65818 | General population: 50.4 ± 12.0, 60.2% | 11 (2014 – 08/2020) (+) | Latent class growth analysis (LCGA) | Anxiety symptoms (4), depressive symptoms (4) | Education, income, socioeconomic status | Multinomial logistic regression model | 55.56% |
| Raina 2021[101] | Canada | Pandemics (COVID-19 pandemic) | 20478 | Middle-aged and older adults from the general population: NR, 15.98% | 4 (2012 – 12/2020) (+) | Latent class growth modeling (LCGM) using PROC TRAJ SAS | Depressive symptoms (3) | Household/family income, living with family/others, social participation, rural region | Latent class growth modeling (multivariable analysis) | 66.67% |
| Reis 2022[159] | Germany | Pandemics (COVID-19 pandemic) | 2203 | General population: 38.63 ± 14.09, 78.2% | 7 (03/2020 – 09/2021) (−) | Latent class growth analyses (LCGA) | Anxiety symptoms (4), depressive symptoms (4), positive mental health (4), stress symptoms (4) | Having a partner, perceived social support, socioeconomic status | R3STEP procedure in Mplus | 77.78% |
| Rosenström 2022[160] | Finland | Pandemics (COVID-19 pandemic) | 4804 | Healthcare workers: 44 ± 11.00, 88.6% | 12 (06/2020 – 05/2021) (−) | Latent class mixed models (LCMM) | General distress (3) | Living with family/others | Multinomial logistic regression model | 61.11% |
| Saunders 2022[161] | UK | Pandemics (COVID-19 pandemic) | 21938 | General population: NR, 76% | ≥6 (03/2020 – 05/2020) (−) | Growth mixture modeling (GMM) | Anxiety symptoms (5), depressive symptoms (4) | Education, household/family income, living with family/others, social participation, rural region | Multinomial logistic regression model | 66.67% |
| Schäfer 2023[49] | Germany | Pandemics (COVID-19 pandemic) | 1275 | General population: 50.06 ± 13.49, 51.5% | 6-7 (03/2020 – 03/2021) (−) | Growth mixture modeling (GMM) | General distress (4), positive mental health (2) | Active coping, coping using emotional support, education, dispositional resilience, hardiness, internal locus of control, optimism, positive reframing, self-efficacy, sense of coherence, sense of mastery, perceived social support | R3STEP procedure in Mplus | 72.22% |
| Shevlin 2023[162] | UK | Pandemics (COVID-19 pandemic) | NR | General population: NR, NR | 5 (03/2020 – 04/2021) (−) | Growth mixture modeling (GMM) | General distress (5), PTSD symptoms (5) | Income, dispositional resilience, internal locus of control, rural region | Multinomial logistic regression model | 66.67% |
| Shilton 2023[163] | USA, Israel | Pandemics (COVID-19 pandemic) | 1362 | General population: 41.02 ± 13.67, 82.5% | 3 (04/2020 – 09/2020) (−) | Growth mixture modeling (GMM) | Anxiety symptoms (4) | Education, income, overall emotion regulation, self-reliance, living with family/others, perceived social support, neighborhood environment | χ² test, Multinominal logistic regression model | 50.00% |
| Skripkauskaite 2023[164] | UK | Pandemics (COVID-19 pandemic) | 5576 | Parents/carers from the general population: 41.20 ± 6.43, 93% | 10 (04/2020 – 01/2021) (−) | Latent class growth mixture modeling (LCGMM) | Anxiety symptoms (3), depressive symptoms (5), stress symptoms (5) | Education, having a partner | Multinomial logistic regression model | 55.56% |
| Welch 2016[165] | USA | Terrorist attacks (9/11 terrorist attacks) | 17062 | Rescue/recovery workers and volunteers, lower Manhattan residents/area workers, and passersby: NR, 52.9% | 3 (2003 –2012) (−) | Group-based trajectory modeling (GBTM) | PTSD symptoms (6) | Education, social integration, household/family income | Logistic regression model | 55.56% |

*Note.* (−), no pre-stressor assessment; (+), pre-stressor assessment; colors represent colors used in Figure 1, grey color = societal level resilience factors, blue color = social level resilience factors, green color = individual level resilience factors

[a] partly overlapping samples from the Building a New Life in Australia (BNLA) study

[b] partly overlapping samples from the The World Trade Center Health Program (WTC-HP)

[c] partly overlapping samples from the UK Household Longitudinal Study

[d] partly overlapping samples from a study on Hurricane Ike

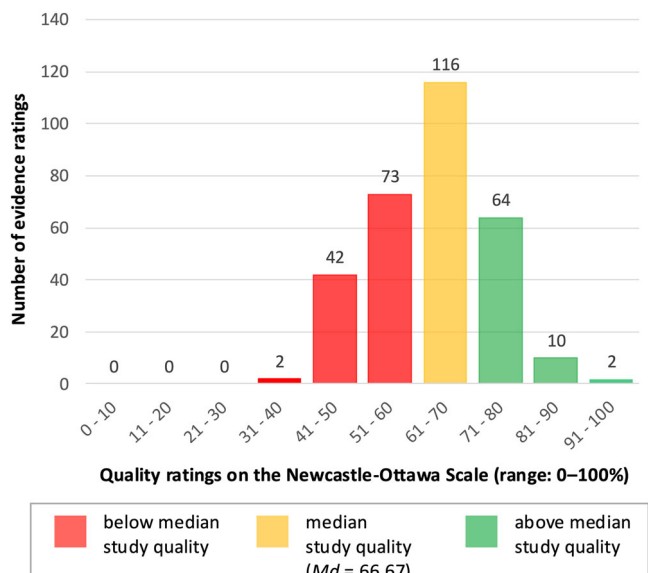

**Fig. 3 | Quality appraisal of included studies and effect estimates.** Distribution of study quality ratings on the modified Newcastle-Ottawa Scale (NOS[73,74]) based on 309 effect estimates included in statistical analyses. Bars reflect the frequency of effect estimates across resilience factors (i.e., circles in Figs. 4 and 5; median values per study and factor were used) with the respective quality rating (e.g., two resilience factors ratings had study quality ratings of 91.0% or higher).

study. Ten studies (20.0%) accounted for uncertainty of class assignments in their regression models.

### Quality appraisal

Overall study quality was moderate (see Fig. 3), with a median study quality rating of 66.67% ($M = 62.17\%$, $SD = 11.23\%$; range: 33.33%–94.44%). Main flaws across primary studies were found for quality of GMM (74.0% high risk), selection (14.0% high risk), followed by outcome assessment (8.0% high risk), and comparability (6.0% high risk). There was no evidence for differences in study quality between studies that assessed either individual, social or societal resilience factors, or a mix of different-level resilience factors, Kruskal–Wallis $H = 4.9$, $p = 0.129$.

### Research question 1: Which societal-level challenges and crises have been examined in OECD member states?

Twenty-nine studies (58.0%) examined pandemics, followed by 9 studies (18.0%) reporting on environmental or natural disasters, and 7 studies (14.0%) investigating terrorist attacks including mass shootings (see Table 2). Involuntary displacements and economic crises were examined in 2 studies each (4.0%), and one study (2.0%) investigated civil unrest. Only 12 studies (24.0%)[90–101] included pre-stressor data, while the remaining 38 studies started during stress exposure up to 40 months after exposure. Studies included 3 to 27 assessment waves ($M = 5.71$ waves, $SD = 4.28$), covering 2 months to 15 years post-stressor.

### Research question 2: What kind of mental health outcomes have been examined to study consequences of those challenges?

All studies used an outcome-oriented approach to resilience, with 16 studies (32.0%) examining posttraumatic stress symptoms, 22 studies (44.0%) assessing depressive symptoms, and 16 studies (32.0%) investigating general distress. Another 14 studies (28.0%) examined anxiety symptoms, and 2 studies (4.0%) assessed stress symptoms. Interestingly, only 5 studies (10.0%) examined positive mental health outcomes (i.e., life satisfaction, mental quality of life, personal or stress-related growth, positive mental health, well-being). Studies identified between 2 and 6 characteristic responses to societal challenges ($M = 3.92$ trajectories, $SD = 0.92$).

### Research question 3: What kind of individual, social and societal resilience factors and mechanisms have been examined?

Thirty-six studies (72.0%) examined the predictive value of individual resilience factors, 43 studies (86.0%) investigated social resilience factors, and only 13 studies (26.0%) assessed societal resilience factors. A variety of 34 individual resilience factors were studied, while 12 social and 8 societal resilience factors were examined in primary studies (see Figs. 4 and 5 for a complete list). Among individual resilience factors, education (28 studies) and income (10 studies) were most often studied. Different types of social support (e.g., perceived/received social support, structural social support; 21 studies) were the most studied social resilience factors, followed by having a partner (16 studies), and living with family/others (15 studies). Among societal resilience factors, most evidence was available for living in rural areas (compared to urban areas; 8 studies) and neighborhood environment (3 studies).

First, we examined associations between the level of resilience factors/mechanisms (individual vs. social vs. societal) and types of societal challenges. We identified a significant link between type of societal challenge and level of resilience factors, Fisher-Freeman-Halton exact tests (FFH) = 42.53, $p = 0.001$, Carmer's $V = 0.30$. There was a focus on societal resilience factors for environmental and natural disasters (i.e., those were examined more often than expected in studies on this stressor type), while many studies on pandemics and terrorist attacks examined individual resilience factors. Social resilience factors were studied equally often across stressor types.

Second, we examined associations between specific resilience factors/mechanisms within one level (e.g., different individual resilience factors) and types of societal challenges. Within individual level resilience factors and mechanisms, there was no evidence for a significant association with stressor types, FFH = 73.67, $p = 0.089$, Carmer's $V = 0.28$. However, there was a trend towards a focus on education for natural disasters, while research on pandemics concentrated on control beliefs, and coping strategies were often examined in the context of terrorist attacks. Also, for social resilience factors the association of single factors with specific stressor types was only close-to-significant, FFH = 48.74, $p = 0.060$, Carmer's $V = 0.26$. However, there was a trend towards a focus on living situations during pandemics, while facets of social support were often studied in the context of terrorist attacks and natural disasters. For societal resilience factors, there was a significant association with stressor type, FFH = 22.88, $p < 0.001$, Carmer's $V = 0.83$, with societal factors being only examined for pandemics and natural disasters. Climate-related factors and collective efficacy were only examined for natural disasters, while aspects of the living environment were more often investigated during pandemics.

### Research question 4: What is the evidence level for each resilience factor and mechanism?

Overall, 478 effect estimates (shown as circles in Figs. 4 and 5) were available for assessing the predictive value of multilevel resilience factors when we compared resilience trajectories to less favorable responses. Of those, 206 pointed to incremental validity of resilience factors above sociodemographic variables and other resilience factors (+++). Five effect estimates showed incremental validity above sociodemographic data (++), and 6 reflected a link with favorable outcomes without control of other variables (+). By contrast, 222 effect estimates suggested no association of resilience factors with resilience trajectories (o to ooo), of those 85 effect estimates trended into a positive direction and 74 effect estimates were numerically negative. A total of 39 effect estimates showed that resilience factors were associated with less favorable responses (- to ---), of which 34 were controlled for sociodemographic data and other resilience factors (---). With respect to effect sizes found for numerically positive effects, the majority of effect estimates (51.7%, 156 out of 302) was very small in size, 30.4% were small, 12.9% were medium, and only 5.0% were large.

At the individual level, the most favorable evidence emerged for individual income, with 12 effect estimates (50.0%) showing incremental validity above sociodemographic variables and other resilience factors (+++), with 3 effect estimates being medium to large (see Fig. 4). Eight

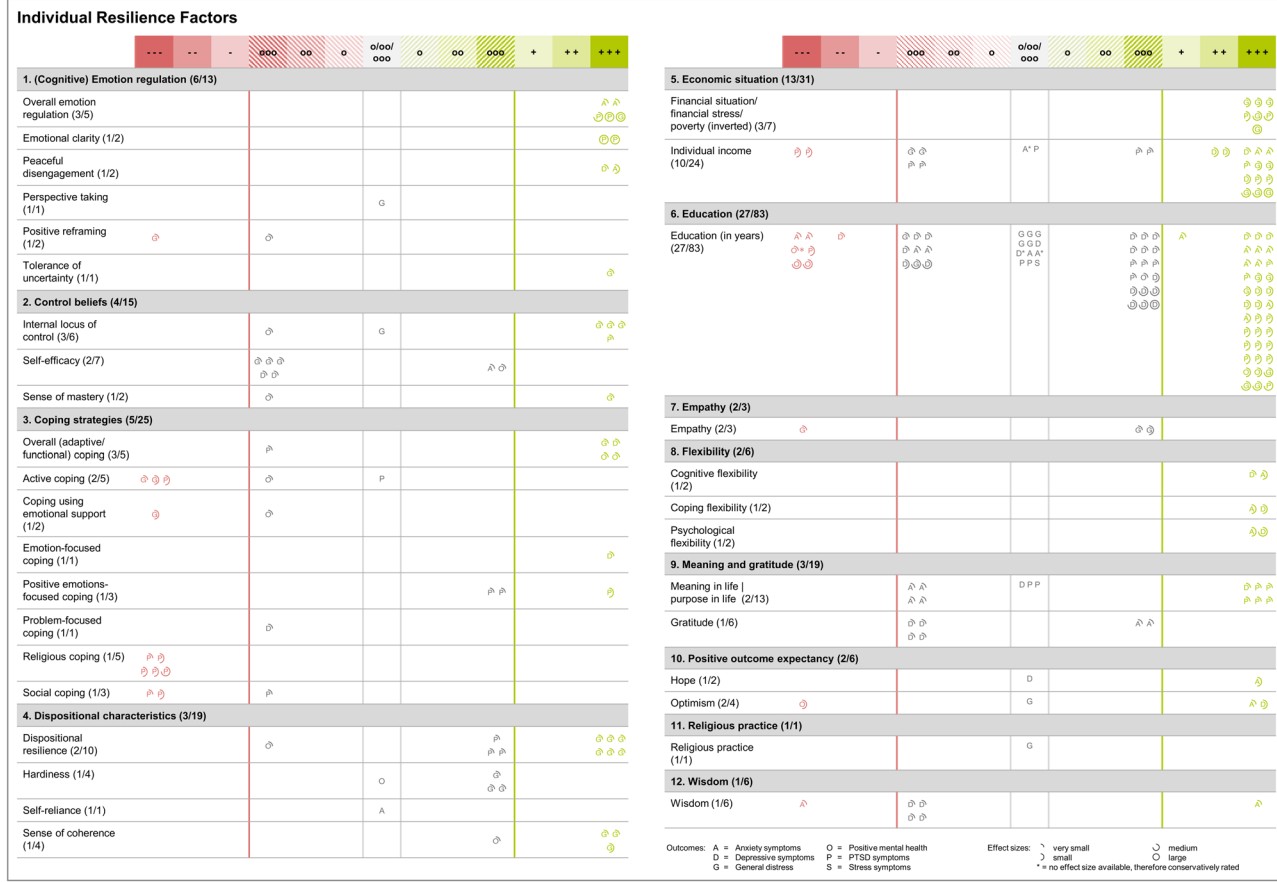

**Fig. 4 | Evidence ratings for individual resilience factors.** Evidence rating per resilience factor for individual resilience factors (both panels) when contrasting resilience trajectories with less favorable responses. Broader categories of resilience factors (e.g., coping strategies, control beliefs) were defined based on previous work on resilience factors[17,19,45,46] and well-established assessments[168,169]; however, categories are not distinct, e.g., positive reframing might also be seen as a type of coping strategy. Numbers in parentheses (e.g., overall emotion regulation [3/5]) indicate the number of studies and effect estimates available for the respective resilience factor (e.g., 3 studies were reporting on overall emotion regulation, with 5 effect estimates being available). Evidence levels range from +++ (= most favorable level of evidence for the respective factor from a single primary study, i.e., the respective factor showed incremental validity beyond other resilience factors and sociodemographic variables with the highest level of control for other variables) to --- (= least favorable level of evidence for the respective factor from a single primary study, i.e., there is evidence for the respective factor being associated with unfavorable trajectories with the highest level of control for other variables). Rating categories o, oo, and ooo represent statistically non-significant findings with different levels of control. Letters in circles indicate types of mental health outcomes (e.g., A = anxiety symptoms) and effect sizes are indicated by pie charts surrounding the respective letter. 25% filling = very small effect; 50% filling = small effect; 75% filling = medium effect; 100% filling = large effect (categories as presented in Table 1 according to Cohen[82]). Details on the rating scheme can be found in Table 1 as well as in the open materials associated with this review (https://osf.io/9xwyu/).

effect estimates suggested no association (o to ooo) with resilience trajectories (33.3%). Similarly, all effect estimates pointed to a favorable effect (7 effect estimates at +++) of low levels of financial stress and poverty, of which 3 were medium to large. Findings for more years spent with education were more mixed with 37 effect estimates suggesting incremental validity (44.6% effect estimates at +++, of those 89.1% were very small or small), but 39 effect estimates yielded null effects (47.0% effect estimates from o to ooo). Evidence for favorable effects was also found for overall emotion regulation abilities, where all effect estimates showed incremental validity beyond sociodemographic variables and resilience factors (5 effect estimates at +++, with 3 effect estimates being medium to large). Very small to small favorable evidence also emerged for peaceful disengagement as a single emotion regulation strategy (2 effect estimates at +++). Small to medium favorable effects were also found for different indicators of flexibility, that is, cognitive flexibility (2 very small to small effect estimates at +++), coping flexibility (2 small effect estimates at +++), and psychological flexibility (2 small to medium effect estimates at +++), with no effect estimate suggesting unfavorable or null effects. However, those resilience factors have only been examined in three studies. While findings were mixed for some resilience

factors (e.g., optimism, positive reframing), effects for active coping (3 very small to small effect estimates at ---), religious coping (5 very small to medium effect estimates at ---), and social coping (2 very small to small effect estimates at ---) pointed into the direction of potentially unfavorable effects, however, also based on a very limited number of studies. Only in one study[50], psychological flexibility was discussed as potential higher-level resilience mechanism.

At the social level, studies focused on living with family or others, perceived social support, and having a partner (see Fig. 5). Evidence for perceived social support was predominantly positive, with 26 (59.1%) effect estimates showing incremental validity beyond sociodemographic variables and other resilience factors (+++). However, 73.1% of those effect estimates were very small to small. Other facets of social support, that is, received social support, and structural social support, were examined less often, yielding mixed results with a comparable number of favorable (+++) and null findings (ooo). For having a partner, evidence was mixed, with only 13 of 45 effect estimates (28.9%) showing significant favorable effects of heterogeneous effect sizes (small to large), while 13.3% of the effect estimates showed very small to small unfavorable effects. For living with family or others, 6 out of 34 effect estimates (17.6%) suggested very small to small

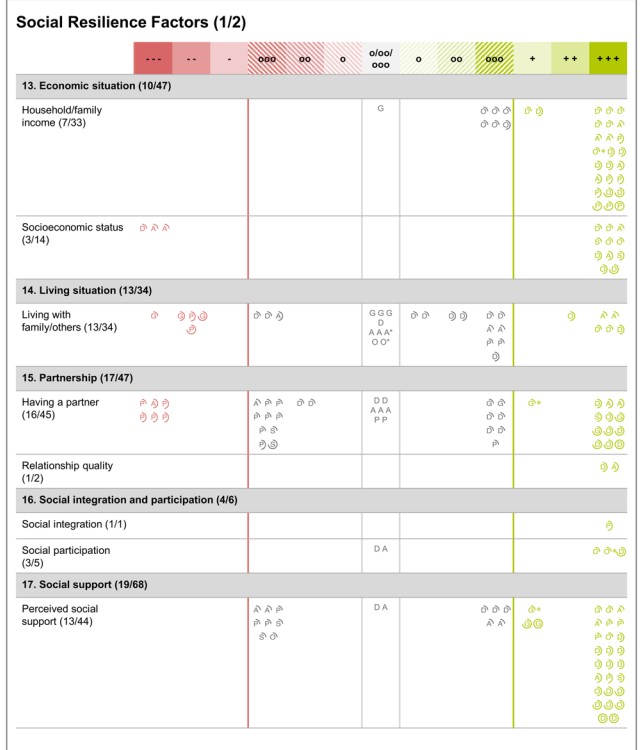

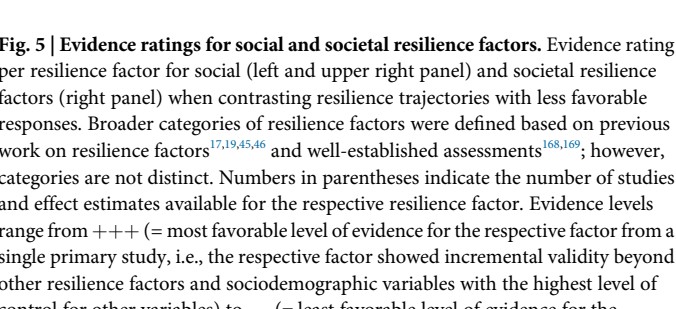

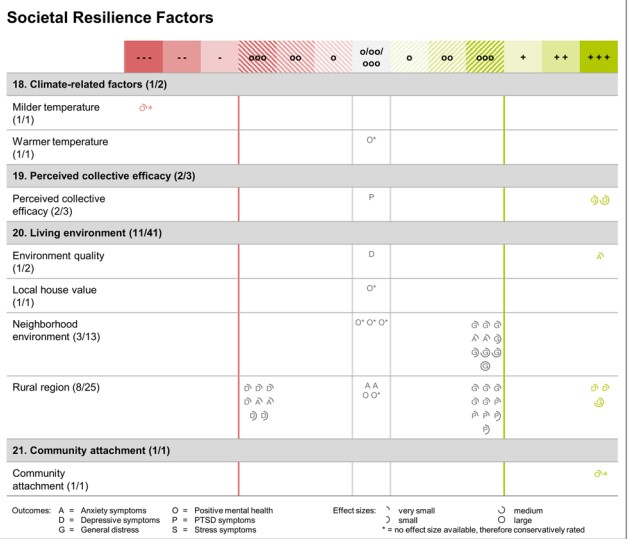

**Fig. 5 | Evidence ratings for social and societal resilience factors.** Evidence rating per resilience factor for social (left and upper right panel) and societal resilience factors (right panel) when contrasting resilience trajectories with less favorable responses. Broader categories of resilience factors were defined based on previous work on resilience factors[17,19,45,46] and well-established assessments[168,169]; however, categories are not distinct. Numbers in parentheses indicate the number of studies and effect estimates available for the respective resilience factor. Evidence levels range from +++ (= most favorable level of evidence for the respective factor from a single primary study, i.e., the respective factor showed incremental validity beyond other resilience factors and sociodemographic variables with the highest level of control for other variables) to --- (= least favorable level of evidence for the respective factor from a single primary study, i.e., there is evidence for the respective factor being associated with unfavorable trajectories with the highest level of control for other variables). Rating categories o, oo, and ooo represent statistically non-significant findings with different levels of control. Letters in circles indicate types of mental health outcomes (e.g., A = anxiety symptoms) and effect sizes are indicated by pie charts surrounding the respective letter. 25% filling = very small effect; 50% filling = small effect; 75% filling = medium effect; 100% filling = large effect (categories as presented in Table 1 according to Cohen[82]). Details on the rating scheme can be found in Table 1 as well as in the open materials associated with this review (https://osf.io/9xwyu/).

favorable effects, while the majority of findings were null effects (67.6%) without a consistent trend into the direction of favorable or unfavorable effects. In line with findings for individual income, there was evidence for significant favorable effects of household income (24 very small to large effect estimates at +++ [72.7% of all effect estimates]), and all but one null finding were trending into the direction of very small to small favorable effects. Socioeconomic status showed a robust link with resilience trajectories (11 very small to medium effect estimates at +++ [78.6%]). None of the studies examined variables that were discussed as social-level resilience mechanisms.

Evidence on the predictive validity on societal-level resilience factors was rare and for most factors limited to single studies (e.g., environment quality, local house value, temperature; see Fig. 5). Aspects of the built and natural living environment were most examined, with only three very small effect estimates (12.0%) showing that living in rural (compared to urban) areas was associated with resilient responses when controlled for sociodemographic variables and other resilience factors, while 22 effect estimates (88.0%) showed no association of living in rural areas with resilience trajectories, with a comparable number of effect estimates trending into the direction of favorable and unfavorable effects. Thirteen effect estimates were available for neighborhood environment, with none of them showing a significant link with resilience trajectories. However, 76.9% of the null effects trended into the direction of favorable effects, with very heterogeneous effect

sizes (very small to large). Perceived collective efficacy showed a link with favorable responses (2 small to medium effect estimates at +++). Resilience mechanisms at a societal level were examined in none of the primary studies.

Subsequently, we examined differences between mental health outcome types. A summary of findings per outcome type is presented in Supplementary Note 6. There was an association between different levels of resilience factors and specific outcome categories, FFH = 27.87, $p < 0.001$, Carmer's V = 0.24. Individual resilience factors were more often examined as predictors of trajectories of general distress and PTSD symptoms, while social resilience factors were most often studied as predictors of anxiety and depressive symptoms, and societal factors were more often examined as predictors of positive mental health outcomes. In general, evidence was relatively rare for some outcome types, ranging from 8 effect estimates for stress symptoms and 47 effect estimates for positive mental health outcomes to 121 for PTSD symptoms. When we examined individual resilience factors across outcomes, favorable effects of individual income remained stable except for PTSD symptoms, where evidence was more mixed (--- to +++); however, higher household income was consistently associated with favorable outcomes for PTSD symptoms with mostly medium to large effect sizes. For education, findings remained inconsistent across outcomes, except for PTSD symptoms, where higher levels of education were consistently associated with more favorable outcomes. Evidence for (cognitive) emotional regulation was limited to very few effect estimates at single-

outcome level. Favorable effects for flexibility were only found for anxiety and depressive symptoms, while other outcome types were not examined. For perceived social support, findings were consistently favorable. Having a partner yielded inconsistent findings across most outcome types, while very small to small negative and null effects (--- to o) were found for PTSD symptoms. Living with a family showed a trend towards favorable responses among most outcomes, while small to medium unfavorable effects were found for general distress and PTSD symptoms. For societal factors, evidence at single-outcome level was very rare, however, consistently favorable effects were found for none of the outcomes.

### Research question 5: What are study, participant and contextual factors impacting on the evidence ratings for resilience factors?

Due to the insufficient number of effect estimates per resilience factor level and a statistical test yielding no significant evidence for between-level differences, Kruskal–Wallis H = 1.24, $p = 0.538$, these analyses were performed combining evidence from different levels of resilience factors. Moreover, analyses were summarized for different outcome types as there was no significant evidence for between-outcome differences in overall evidence ratings, Kruskal–Wallis H = 6.13, $p = 0.293$.

First, we examined aspects of study design. There was no statistically significant evidence for differences between evidence ratings derived from studies with and without pre-stressor data, U = 4529.50, $p = 0.842$. Also, for the number of assessment waves in total, $r_s = 0.01$, 95% CI [−0.11, 0.12], $p = 0.925$, the number of assessments after stressor exposure, $r_s = 0.00$, 95% CI [−0.12, 0.13], $p = 0.966$, and the number of trajectories identified by means of GMM, $r_s = 0.04$, 95% CI [−0.08, 0.16], $p = 0.509$, we found no statistically significant evidence for an association with effect estimates. There was no statistically significant evidence for an association between the total sample size at baseline and evidence ratings, $r_s = 0.02$, 95% CI [−0.11, 0.13], $p = 0.749$. Evidence ratings were less favorable when a larger number of variables was included in logistic regression models, $r_s = −0.22$, 95% CI [−0.33, −0.10], $p < 0.001$, and more resilience factors were simultaneously examined, $r_s = −0.25$, 95% CI [−0.36, −0.13], $p < 0.001$.

Second, we analyzed associations with sample characteristics. Evidence ratings were more favorable when a larger proportion of female participants was examined, $r_s = 0.21$, 95% CI [0.09, 0.33], $p < 0.001$, while older average age of the sample was related to less favorable ratings for resilience factors, $r_s = −0.21$, 95% CI [−0.34, −0.08], $p = 0.002$. There was no statistically significant evidence for differences between representative or non-representative samples, U = 6397.50, $p = 0.161$.

Third, we examined differences in evidence ratings between specific stressor types, without finding statistically significant evidence for differences between different types of societal challenges, Kruskal–Wallis H = 6.94, $p = 0.225$.

### Evidence from recovery versus less favorable trajectories

Evidence ratings for the comparison of recovery trajectories versus less favorable responses are presented in Supplementary Note 7. The number of effect estimates was smaller (81 effect estimates), of which only 12 ( + + +, 14.8%) suggested incremental validity of resilience factors beyond sociodemographic variables and other resilience factors. Another 34 effect estimates (42.0%) were non-significant but showed a trend towards favorable effects. In general, findings were less consistent and suggested more null effects for resilience factors including those that showed favorable effects for the comparison between resilience trajectories and less favorable responses (e.g., individual and household income, perceived social support). Evidence for other individual and social resilience factors (e.g., flexibility and socioeconomic status) as well as societal factors was almost absent.

### Sensitivity analyses

There was no statistically significant evidence for an association of study quality and evidence ratings, $r_s = 0.06$, 95% CI [−0.06, 0.17], $p = 0.357$. Moreover, we examined aspects of timing: There was no statistically significant evidence for associations between evidence ratings and the

time between the last pre-stressor assessment and occurrence of the stressor, $r_s = −0.10$, 95% CI [−0.24, 0.04], $p = 0.131$, the time between stressor exposure and the first post-stressor assessment, $r_s = −0.03$, 95% CI [−0.17, 0.10], $p = 0.579$, and the time period between stressor and last assessment, $r_s = −0.03$, 95% CI [−0.15, 0.09], $p = 0.601$.

## Discussion

This systematic review examined the predictive validity of multilevel resilience factors for mental responses to societal challenges in OECD member states. We identified 50 studies that examined responses to multiple societal challenges, that is, pandemics, environmental or natural disasters, terrorist attacks, involuntary displacement, economic crises, and civil unrest. Overall, 54 resilience factors were examined, of which 34 were individual factors, 12 were social and 8 societal resilience factors, with a special focus on societal resilience factors in studies on environmental and natural disasters, while individual resilience factors had a greater chance to be included in studies on pandemics and terrorist attacks. The most favorable effects were found for individual income and low financial stress, (cognitive) emotion regulation and facets of psychological flexibility at the individual level, for perceived social support, socioeconomic status, household income, and relationship quality at the social level, and for environmental quality at the societal level. We found more favorable effects of resilience factors in samples comprising more women and younger participants. For many other well-established resilience factors (such as self-efficacy, locus of control, positive reframing, and optimism), findings were mixed with a comparable number of favorable and null effects. For active coping, religious coping, social coping, and milder temperature, there was also evidence for an association with unfavorable responses. Research into social resilience factors – beyond social support – and societal resilience factors was rare and research into higher-order resilience mechanisms was almost absent.

In general, even among resilience factors showing favorable effects, 86.9% of the effect sizes were very small to small and only 1.9% were large. On the one hand, this is in line with previous claims[22,25,69] that resilience factors often only show small associations with mental responses to stress. On the other hand, this finding is also a by-product of our methodological approach as we were specifically interested in the incremental validity of resilience factors beyond other variables. Therefore, whenever possible, we used data from models including the largest number of control variables, which inherently reduces the exploratory value of single predictors[102]. This effect was also present in studies comparing single predictor models with models including a larger number of predictors[90]. In line with this evidence from primary studies, our analyses on methodological characteristics yielded that a larger number of variables and resilience factors in regression models were associated with less favorable evidence ratings per factor. Thus, our results show that resilience factors across all levels had mostly small incremental validity above sociodemographic data and other resilience factors. Future studies will have to examine whether this reflects their overall low exploratory value[22,25,69] or rather shows that the most important information available from resilience factors is what they share with other resilience factors[47,49]. Interestingly, for many factors, effect sizes were highly heterogeneous across studies ranging from very small to large effects. These differences may originate from the inclusion of a varying number of variables in regression models but can also reflect that contextual factors modulate the importance of resilience factors as this is suggested in flexibility frameworks[25,69,70].

Looking at single levels, individual resilience factors were most often examined with the largest number of studies reporting effect estimates for sociodemographic characteristics that could be viewed as resilience factors (i.e., education, individual income). The strongest evidence emerged for individual income and low financial stress, which may again underline the relevance of economic security for mental health in face of any societal challenge[103]. While a large number of psychological variables were examined as individual resilience factors, the number of effect estimates per factor was small, suggesting that there is still little consensus on individual psychological resilience factors[25,49], and many studies only examine a small number of

factors. Most favorable findings for psychological variables emerged for different aspects of cognitive emotion regulation[104] and flexibility[24], yet with partly very small and small effect sizes. Evidence for many other psychological variables was either weak (e.g., hardiness, locus of control, self-efficacy, self-reliance, sense of coherence, meaning and gratitude), highly inconsistent including associations with favorable and unfavorable mental responses as well as null effects (e.g., optimism, wisdom), or consistently pointed to null or unfavorable effects (e.g., active coping, coping using emotional support, social and religious coping). These findings may support the idea of flexible emotion regulation[105] that a match between coping resources and resilience factors on the one hand and situational demands, on the other hand, might be key for successful coping[25]. For example, active coping might be helpful when situations allow to find flexible solutions, however, in situations where one's own scope of action is limited, a general preference for active coping might even be harmful as it might hinder the use of more useful strategies (e.g., acceptance, reappraisal)[83]. The same may apply to coping using emotional support, which might be helpful when such support is available from family and peers, but might be unfavorable when the stressor itself impacts social resources as this was the case during the COVID-19 pandemic. The concept of regulatory flexibility[24] ties in with this idea of a match between resilience factors and situational demands, and our favorable findings for different aspects of flexibility may preliminarily support the importance of a flexible selection of resources used for coping with societal challenges. However, effect sizes found for flexibility were mostly very small to small and the broader concept of regulatory flexibility[24] has not been examined in primary studies. Included studies only focused on single components of flexibility (i.e., cognitive flexibility[26], coping flexibility[27], psychological flexibility[50]). Studies examining all components of regulatory flexibility (i.e., flexibility mindset, flexibility sequence[70]) are needed.

At the social level, research very much focused on (perceived) social support. This might also reflect that societal challenges often impact on social relationships making them a focus of research[106]. In line with previous reviews on social support[18,52], we found mostly favorable effects of perceived social support with the majority of studies reporting very small to small positive effects. For received social support null findings outweighed favorable effects, while there was a trend towards beneficial effects for structural social support, however, based on a very limited number of studies. This may suggest that the effects of different aspects of social support vary depending on contextual factors, which may also include resilience factors at a societal level (e.g., macro-economic situations, income [in]equality, cultural dimensions, access to natural spaces, trust in institutions). By contrast, findings on having a partner and living with family (or others) were highly mixed, some effect estimates suggested a link with resilient responses, while others yielded null or even negative effects. Future studies need to shed light on these contextual factors potentially modulating effects of social resilience factors and may also examine the interplay between one's desire for social support, its source, availability and provision[107].

Evidence was the weakest for societal resilience factors, which had only been examined in the context of natural disasters and pandemics. Most favorable evidence emerged for environmental quality and collective efficacy, that is, the belief that actions by a societal group impact their shared future[108]. Living in rural (compared to urban) areas and neighborhood quality were examined in the largest number of studies but showed almost consistently no significant association with mental responses. However, in case of neighborhood quality, there was a trend towards a link to favorable responses. The study of other societal-level resilience factors (e.g., income inequality, efficacy of crisis communication) might be negatively impacted by the often nationally funded research projects as differences might rather occur across nations than between individuals living within one society. Thus, studies solely conducted in one country often examining a single nation might be insufficient to shed light on these factors due to within-country variance restrictions. This underlines the need for sharing forces in face of societal challenges to allow to study not only individual factors associated with resilient responses, but to also focus on factors that might lie on a societal level.

In contrast to consistent calls for more research into resilience mechanisms[11,14,22,109,110], such research is still almost absent, with none of the included primary studies explicitly focusing on resilience mechanisms. Research into different aspects of flexibility[26,27,50] might be interpreted as such a higher-level individual resilience mechanism (but see Kalisch et al.[23] for a critical reflection on flexibility as a resilience factor vs. strategy-to-situation fitting as a resilience mechanism), while research into social and societal resilience mechanisms is missing in studies on individual responses to societal challenges. Our finding of between-study and between-outcome inconsistencies for single resilience factors at all levels, that is, single resilience factors were important in one study, but not in another; or were important for one outcome, but not for another, support the idea that such mechanisms might be of greater exploratory power compared to research solely focusing on an ever-growing number of single factors. This claim is further supported by the often small and heterogeneous effect sizes and low exploratory value of resilience factors across all levels. In line with previous findings[17,25], even when a larger number of resilience factors was examined, their sum did not account for the complex phenomena of resilient outcomes.

Our review identified research into social and societal resilience mechanisms as one of the most important evidence gaps. On a social level, one may think of mechanisms relevant to establishing and maintaining social relationships, which in turn lead to a general feeling of connectedness[111]. On a societal level, the mobilization of resources (e.g., facilitating communication between stakeholders), self-regulation processes (e.g., capabilities to make decisions in times of crises) and capacities to transform societal systems (i.e., the ability to learn from previous challenges) might be viewed as potential resilience mechanisms[31]. Future research will benefit from integrating knowledge from other fields examining societal adaptation processes (e.g., sociology, security research) and from adapting a multilevel and multisystemic perspective[14], and might start with disentangling resilience factors and mechanisms at a conceptual level. Such conceptual clarity will help to examine the complex interplay between resilience factors and mechanisms at different levels.

## Limitations

Despite the strength of this systematic review, our findings need to be interpreted in light of their limitations. First, the review summarizes evidence on responses to societal stressors (e.g., pandemics, wars, and armed conflicts). For the identification of these stressors, we used a recently published list of public health disasters[2], however, a definite typology of stressors is still missing, with some reviews using other classifications[112]. Thus, the use of a specific list of stressors increased interrater reliability but might have biased our results. Second, our analyses relied on longitudinal data, yet associations of resilience factors reflect (partial) correlations and do not allow for conclusions on a causal link between resilience factors and mental responses nor did we examine prediction of resilient responses. Thus, our findings should not be misinterpreted as evidence for a core set of resilience factors and mechanisms that should be targeted in resilience interventions. Such intervention targets should be derived from studies using more complex designs to investigate the interplay and potentially causal links between resilience factors/mechanisms and mental responses over time[113,114]. Third, we decided to include studies using GMM. This approach was chosen as it is the most common approach in resilience research[32,33], it allows for sufficient between-study comparability, and the criticism of GMM is not directly related to multinomial logistic regression models, which were the focus of our analyses. However, the use of GMM is not without criticism[67,115] (see also Supplementary Note 1), with models being criticized for artificially producing inflated prevalence rates for resilient responses by being highly constrained[115,116]. Moreover, a lack of pre-stressor data might also result in inflated prevalence estimates for resilient responses

as initial increases in distress might be missed resulting in recovery responses being mis-labeled as resilience trajectories. Yet, previous reviews[17,33] did not provide evidence for such a bias. Inflated prevalence rates for resilient responses may also impact on our evidence ratings for resilience factors and might increase random error, which could have induced bias in both directions—important factors might be missed, or the importance of resilience factors might be overestimated. Fourth, we developed a rating scheme to compare evidence levels for different resilience factors. This approach has been chosen as standard meta-analysis on odds ratios was not applicable due to large between-study differences in logistic regression models (i.e., with respect to the number and type of predictors). In line with recommendations of the Cochrane Collaboration[117], we used a version of vote counting, which was enriched by information from statistical significance tests[76] and effect sizes. However, using this approach, our results are impacted by the limitations of statistical significance testing[80]. Future systematic reviews based on more homogeneous studies may use meta-analyses to obtain meta-analytical effect estimates allowing for conclusions on effect sizes. As we were unable to perform meta-analysis, also standard methods to assess a potential publication bias (i.e., the greater likelihood of significant results to be published[118]) were not applicable. On the one hand, in most primary studies, the predictive value of the respective resilience factor was not the main research focus, which limits the potential impact of a publication bias. On the other hand, we found a large number of significant, yet very small effect estimates, which may point to a potential publication bias. Future systematic reviews using meta-analysis should examine such a bias by means of statistical methods[119].

Other limitations derived directly from the included primary studies. We were unable to run proper analyses on between-outcome differences as the number of effect estimates per resilience factor and outcome was too small. Conclusions on 30 resilience factors were based on findings from single studies examining single societal challenges, which limits their generalizability. Moreover, we were not able to derive recommendations for single types of societal challenges as the number of effect estimates per stressor type was too small. Our analyses on between-stressor differences showed that there were links between different levels of resilience factors and specific stressor types with a focus on societal resilience factors in face of environmental and natural disasters and a spotlight on individual resilience factors in the context of pandemics. We found no evidence for between-stressor differences in evidence ratings; however, this finding may also be accounted for by a low number of studies for specific stressor types (e.g., economic crises, civil unrest). Moreover, specific stressors had been repeatedly examined – for example 100% of the studies on pandemics examined the COVID-19 pandemic and all studies on terrorist attacks examined the mental responses to the September 11 attacks (9/11). Thus, future studies need to examine whether our findings also hold for other stressors falling into the same larger category and whether resilience factors might only be beneficial for some types of stressors but not for others. These studies will also provide empirical insights into whether the selection of stressors used for this review[2] represents a sufficiently homogeneous subgroup or needs further refinement. Such a typology of stressors moving beyond traumatic stress[120,121] is urgently needed and may also help to classify stressors along important dimensions (e.g., intensity, predictability, controllability, novelty, duration[122]). Also, timing of post-stressor assessments varied substantially between studies, with some studies' assessments starting in the first hours after exposure and others after 3.3 years[123]. Also the number of post-stressor assessments (3 to 27 assessments) and the length of follow-up varied substantially from 1.7 months to 14 years, which might also impact on the relative importance of resilience factors[52,114]. Another limitation of our review is missing pre-stressor data with only 24% of the studies including pre-stressor assessments. Such studies are particularly challenging in face of societal stressors, which often have a sudden and unforeseen onset such as terrorist attacks or the COVID-19 pandemic, preventing the collection of pre-stressor data. At the same time, those studies are needed as a lack of pre-stressor assessment may result in underestimated prevalence rates for recovery responses (as initial increases of distress have been missed)[17]. Moreover, resilience factors that were assessed after stressor exposure might already be impacted by the stressor itself or represent correlates of stable between-person differences in mental health unrelated to responses to a specific stressor[49]. We examined by means of sensitivity analyses whether the availability of pre-stressor data impacted evidence ratings, without finding statistically significant evidence for differences between studies with and without pre-stressor data. Yet, we cannot exclude that these studies have introduced biases in our evidence ratings. Other problems might be caused by missing representativeness, with most studies using convenience samples, and low diversity of study samples. In general, defining a target population for studies on consequences of stressor exposure is challenging. For some stressor types (e.g., environmental and natural disasters, terrorist attacks), one may ideally recruit a representative sample of those living in a respective area or being present during a specific event at a given time, for other stressors (e.g., pandemics) such approaches might be less suitable as between-individual heterogeneity in exposure levels is larger. Such challenges can be addressed by accounting for levels of exposure in statistical analyses[23], yet this has not been sufficiently done in many studies. Especially for those studies using convenience samples, we cannot exclude that those who were affected the most by stressors (e.g., minority groups[124]) were not sufficiently healthy to participate in the studies included in our review, which could have resulted in substantially biased findings. Consequently, we cannot conclude that those resilience factors identified as important resources are equally important to all people exposed to a specific stressor in OECD member states.

## Implications for future research

Our review forms a base for future research into resilience factors. So far, studies that examine a broad range of psychological resilience factors are rare[27,49–51], with most studies investigating incremental validity only beyond income, education, or socioeconomic status. However, studies examining a larger number of psychological variables are needed, and resilience research may benefit from collaborative effort to define and regularly assess a core set of most promising resilience factors at different levels. Such effort should result in large-scale international mental health surveillance projects that monitor public mental health in face of societal challenges[7]. Data from such projects might also allow for valuable between-country comparisons, which will enable research on so far understudied societal factors (e.g., income [in]equality, gender [in]equality, cultural dimensions). Our findings suggest that individual income and low financial stress, (cognitive) emotion regulation, and aspects of flexibility are most promising at the individual level, while household income, socioeconomic status, perceived social support, and relationship quality require more intense research at the social level, and collective efficacy as well as environmental quality should be further examined at the societal level. Moreover, research is needed that identifies the boundary conditions that modulate the effects of resilience factors showing mixed evidence (e.g., having a partner, living with others, education, wisdom). A larger number of high-quality international panels will allow for integrating data into a large-scale dataset suitable for individual participant meta-analysis[125], which will help to shed light on participant-level modulators (e.g., age and gender). Our review suggested that the importance of resilience factors might be larger for women and younger populations, however, our database did not allow for an in-depth analysis on gender- and age-related differences for resilience factor levels or single factors. Moreover, such studies may also make use of advanced methods for predictor selection (e.g., machine learning approaches[126], LASSO regression[27]), which provide knowledge on the relative importance of resilience factors moving beyond statistical significance[80]. By using more complex analyses, those studies may also allow to identify dynamics of the importance of resilience factors over time, which are not captured by our review. For example, in a recent study[113], the importance of perceived social support was found to vary in the first year after stressor exposure and also between different sources of social support.

On the long run, high-quality international panels will inform the development, improvement, and evaluation of prevention and resilience

programs. So far, most resilience programs focus on the individual and aim at strengthening a large set of individual resilience factors[13,19], implicitly transferring responsibility for resilience to individuals rather than groups or societies. Our multilevel approach to resilience factors might have the potential to guide the development and evaluation of future programs as those may be designed to address resilience factors at multiple levels[127]. Constant multidimensional measurement of central cross-situational resilience factors on a national level can help guide public promotion or intervention efforts. At the same time, program evaluations should move beyond measuring changes in mental health and psychological resources[19,128]. Future studies should also derive recommendations for so far neglected suitable outcomes at the social and societal level[129]. Holistic programs might employ a multilevel approach to all components of the process: resilience factors, resilience mechanisms and outcomes. Initial initiatives in this direction were outlined and discussed, particularly during the COVID-19 pandemic as a global stressor, in relation to crises preparedness and responsiveness at various levels (e.g., individual and community resilience)[130,131]. The studies summarized in this review could be a useful extension for possible starting points for such multilevel evaluations where event-related resilience interventions could be examined as predictors of post-stressor changes in multidimensional outcomes.

## Conclusion

This review examined the predictive value of individual, social and societal resilience factors for mental responses to societal challenges and crises. We found a focus on individual resilience factors in research, while social and societal resilience factors have been examined less often. Among the resilience factors examined in our review, there was no single factor outshining all others. We found evidence for higher income and socioeconomic status, lower financial stress, better cognitive emotion regulation, higher perceived social support and higher levels of multi-faceted flexibility being associated with resilient stress responses. However, the majority of effect estimates for incremental validity of resilience factors were very small to small. For many resilience factors – including self-efficacy, education, and optimism – findings were mixed suggesting that the fit between resilience factors and situational demands might be key to understand the complex phenomenon of successful adaptation. Future large-scale international studies on public mental health should include pre-stressor data, examine a larger number of resilience factors, should focus on social and societal resilience factors, invest in the straight-forward study of resilience mechanisms at multiple levels, and employ more statistical modeling approaches suitable to capture complex temporal dynamics and between-factor interactions.

## Data availability

Data and materials used for this review are accessible at the Open Science Framework (OSF): https://osf.io/9xwyu/.

## Code availability

Code for analysis is available at the Open Science Framework (OSF): https://osf.io/9xwyu/.

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

## Acknowledgements

This research was funded by the Robert Koch Institute, Berlin, Germany (Project code: LIR_2023_01). The funder had no role in study design, data collection and analysis, decision to publish or preparation of the manuscript. Two co-authors, CC and CK, are researchers at the Robert Koch Institute and were involved in the study design, data analyses and preparation of the manuscript, with no personal or financial conflict of interest being present in this process. The research question was defined in a call for tenders by the Robert Koch Institute, for which the co-authors SKS and KL submitted an offer that was selected by the funder. We thank Prof. Dr. Kurt Hahlweg and Prof. Dr. Wolfgang Schulz as well as the group of Caroline Jung-Sievers for the fruitful discussion on social and societal resilience factors. Our evidence rating scheme was adapted and improved based on the helpful comments of one anonymous reviewer. We thank this reviewer for their time and thoughts dedicated to our work. Moreover, we thank Florence Herian, Franziska Schütz, Isabelle Weber, and Desirée Wild for their help in preparing this review.

## Author contributions

Sarah K. Schäfer: conceptualization (lead), formal analysis (lead), methodology (lead), writing – original draft preparation, funding acquisition (equal); Max Supke: conceptualization (supporting), data curation (lead), formal analysis (supporting), methodology (supporting), writing – review and editing (equal), project administration (lead); Corinna Kausmann: conceptualization (supporting), methodology (supporting), writing – review and editing (equal); Lea M. Schaubruch: conceptualization (supporting), data curation (lead), formal analysis (supporting), methodology (supporting), writing – review and editing (equal), project administration (supporting); Klaus Lieb: resources (supporting), supervision (equal), writing – review and editing (equal), funding acquisition (equal); Caroline Cohrdes: resources (lead), supervision (equal), writing – review and editing (equal).

## Funding

## Competing interests

The authors declare no competing interests.
