## [Peer Review File · Communications Psychology]

5th Jan 24

Dear Professor Schäfer,

Thank you for your patience during the peer-review process. Your manuscript titled "Individual, social and societal resilience factors in the face of societal challenges and crises: A systematic review of adult populations from OECD member states" has now been seen by 2 reviewers, whose comments are appended below. You will see that they find your work of some potential interest. However, they have raised substantial concerns that must be addressed. We are interested in the possibility of publishing your study in *Communications Psychology*, but would like to consider your responses to these concerns and assess a revised manuscript before we make a final decision on publication.

We therefore invite you to revise and resubmit your manuscript, along with a point-by-point response to the reviewers. Please highlight all changes in the manuscript text file.

Editorially, we consider it important that the revised manuscript includes a meta-analytic analysis of the included studies' effect sizes where there is sufficient data to allow for it. We also require an estimate or correction for publication bias.

Please also include additional methodological detail regarding data extraction and analytic strategy. The rationale for focusing on growth mixture modeling should be further clarified. The limitations and caveats of the findings should be thoroughly discussed. The discussion should focus on the policies and interventions tested in the analyzed corpus of studies.

If the revision process takes significantly longer than five months, we will be happy to reconsider your paper at a later date, provided it still presents a significant contribution to the literature at that stage.

Please use the following link to submit your revised manuscript, point-by-point response to the Reviewers' comments with a list of your changes to the manuscript text (which should be in a separate document to any cover letter) and any completed checklist:

[link redacted]

Please do not hesitate to contact me if you have any questions or would like to discuss the required revisions further. Thank you for the opportunity to review your work.

Best regards,

Jennifer Bellingtier

Jennifer Bellingtier, PhD

Senior Editor

Communications Psychology

EDITORIAL POLICIES AND FORMATTING

We ask that you ensure your manuscript complies with our editorial policies. Please ensure that the formatting requirements are met, and any checklist relevant to your research is completed and uploaded

as a Related Manuscript file type with the revised article. I have attached a detailed table to help with this process.

* TRANSPARENT PEER REVIEW: Communications Psychology uses a transparent peer review system. This means that we publish the editorial decision letters including Reviewers' comments to the authors and the author rebuttal letters online as a supplementary peer review file. However, on author request, confidential information and data can be removed from the published reviewer reports and rebuttal letters prior to publication. If your manuscript has been previously reviewed at another journal, those Reviewers' comments would not form part of the published peer review file.

REVIEWER EXPERTISE:

Reviewer #1 Stress, resilience, societal health and well-being

Reviewer #2 Stress, resilience, societal health and well-being

Reviewer #1 (Remarks to the Author):

I have read the manuscript with great interest and was impressed by the huge amount of work that was done. The authors were able to identify a large number of studies assessing trajectories of mental health problems following stressful events/societal changes, varying from the 9/11 attacks, financial crisis to the COVID-19 pandemic, all using GGM or similar statistical techniques, and should be complimented for this accomplishment.

Because of my enthusiasm and high esteem for their efforts, I very sorry to say that in my view the current manuscript does not really address the main research questions 4 and 5. I hope that I can explain my main and major problem with the manuscript that is based on their extensive research.

Table 1 (pag. 10) provides an overview of the criteria used to rate the evidence levels of the outcomes of include studies. The tables shows that, as long as a resilience factor is significantly associated with resilient outcomes, the rating is positive (+ to +++): studies without control or other variables were rated with a +, studies with sociodemographic control variables with ++, and studies with sociodemographic

control variables and other resilient factors with ++++. The main and major problem of this strategy is that it tells the reader little to nothing about the relevance of the regarding resilience factors: significance level is a very poor and unacceptable measure for the evidence level and should be replaced with well-established effect size measures across studies (see meta-analysis). Trivial or minimal and thereby irrelevant associations can easily be significant when the sample is large enough (the social science literature is full of such associations). It is therefore impossible to offer critical and helpful comments on large parts of the results (and discussion) section that were primary based on the evidence ratings (the unclear distinction between resilience and sociodemographic factor is a minor problem in this perspective). In the results section the authors also provided an overview of factors that were more or less examined in the included studies, but I do not know if such an overview is of interest for the readers because the authors excluded all studies not using GGM or comparable methods (see further below).

Other problems I encountered were:

1. The included studies examined trajectories of mental health by means of GMM (or comparable methodological approaches) aiming at identifying different patterns of mental health over time) and investigated individual, social and/or societal resilience factors as their predictors (i.e., as an independent variable in a regression analysis). Importantly, Table 2 shows that included studies differed strongly in timing of the assessments: part of the studies included pre-event/societal change assessments while other studies started one or more years of the event/societal changes and/or differed in timing of subsequent surveys. Part of the studies conducted the last assessment within 1-2 years after the event/societal changes, while other conducted the last assessment several years later. In other words, totally different study designs were applied in the included studies. How these important differences were accounted for in the analyses (including weighing of evidence levels) is not explained nor clarified, as if timing/time lags do not matter. I tried to understand it, but eventually did not make sense to me.

2. In addition, the type of events/societal changes the included studies focused on differ strongly. The majority focused on the COVID-19 pandemic and others on specific events like the 9/11 attacks. Please clarify why you believe that the outcomes if these studies can be combined and not separated by type of event. I could not discern much similarities between (for example) these two type of events.

3. I understand that the authors used the term resilience as an outcome or label: the maintenance or fast recovery of mental health during or after stressor exposure. In many studies resilience is, in brief, defined as a personal capacity "to bounce back" during or after stressful events or circumstances. It could help the reader if the authors explained why they choose the "outcome" definition. In addition, if no (non-retrospective) pre-event mental health data are available in the included studies, what do post-event trajectories actually say? For instance, if people already suffered from pre-existing severe mental health problems and belong to the so-called chronic trajectory, are they then not resilient? Please explain. In the trajectories figure on page 44 a sub-class of the so-called resilience trajectory is presented where mental health increase after the event/societal changes but decrease (so it seems)

relatively fast. Please explain after how many days/weeks/month initial mental health problems must decrease to define this group as resilient to be able to compare studies.

4. But more importantly, the authors would help the reader much more by explaining the advantage of assessing factors associated with trajectories as assessed with GMM above assessing factors associated with the development of mental health problems at different moments after the event/societal changes in the aftermath (that certainly enables classifications of trajectories of for instance PTSD too). Do they not inform us about individual, social; and social risk factor at different moment after the event/societal changes? I do not think so.

5. GGM and latent class analyses are very interesting techniques but not without limitations. There is always classification error and studies differ in the amount of classification error. As far as I could see in the materials I received, this topic was (much to my surprise) ignored in the quality assessment of the included studies as well as in the evidence ratings. With respect to the quality assessment, the authors wrote that “Only a small share (11 studies) was representative of the respective target population with most using convenience samples”. A well-known major problem with convenience samples is that we simply do not know whether the results are representative for a certain population or not. Please explain or clarify why it is justified to include studies with an unknown representativeness. In my personal view, such studies should be excluded because the quality rating is by definition (very) low.

As said above, I am very sorry to say that in my view the main research questions 4 and 5 are not really answered because of the chosen analytic strategy. At this moment, I cannot oversee the relevance of manuscript that answers the research question 1-3 only. Nevertheless, the authors created a large impressive database of studies. I hope that the authors continue with their important work and that my review is of help to a certain extent.

Reviewer #2 (Remarks to the Author):

Overall Evaluation

This manuscript presents a systematic review examining the predictive validity of individual, social, and societal resilience factors in maintaining mental health trajectories during major societal crises. The topic is highly significant given recent events highlighting the need to understand public resilience. Key strengths of the review include the longitudinal, multi-level approach, examination of incremental validity, and focus on understudied societal factors. The review makes an important contribution in

consolidating current evidence across crises and identifying critical gaps to guide future research. However, enhancements to the methodology, context, and implications would further strengthen the quality and impact of the findings.

Major Revisions

1. Provide more background on hypothesized resilience mechanisms that may mediate between factors and outcomes. Elaborate on what these mechanisms may be and how they operate at multiple systemic levels.
2. Summarize differences/similarities in resilience patterns and factors across the societal crises, both from current evidence and informed speculation based on crisis characteristics.
3. Discuss in more depth the policy, practice and crisis preparedness implications beyond informing resilience interventions. Particularly relevant are building societal resilience capabilities.
4. Enhance methodological detail and clarity throughout, particularly around data extraction, analysis procedures, and growth mixture modeling rationale.
5. Acknowledge limitations with the longitudinal observational design, sample diversity, and consistency of data quality over time.
6. Refine categorization of resilience factors, clarify their derivation, and discuss their theoretical grounding.
7. Explore the potential for more complex statistical modeling of the relationships between resilience factors and mental health trajectories.

Addressing these areas will further enhance the quality, rigor and potential impact of this timely review synthesizing the current understanding and future directions of resilience research across systemic levels and societal shocks.

Reviewer #1:

1. I have read the manuscript with great interest and was impressed by the huge amount of work that was done. The authors were able to identify a large number of studies assessing trajectories of mental health problems following stressful events/societal changes, varying from the 9/11 attacks, financial crisis to the COVID-19 pandemic, all using GGM or similar statistical techniques, and should be complimented for this accomplishment.

Because of my enthusiasm and high esteem for their efforts, I very sorry to say that in my view the current manuscript does not really address the main research questions 4 and 5. I hope that I can explain my main and major problem with the manuscript that is based on their extensive research.

Response: Many thanks for your time dedicated to our manuscript. We highly appreciate your critical comments and concerns related to our review approach. These helped us a lot to prepare a revised version of our evidence rating, which addresses your concerns. We now included information on effect sizes in our rating to not solely rely on statistical significance. Also in this version of our rating, we cannot overcome all limitations of our vote counting approach. However, we still believe that such an approach is superior to a solely qualitative summary, which is at risk for several biases due to selective reporting. To our knowledge, there is also no review available summarizing the evidence for a broad range of resilience factors. So, we view our systematic review as a starting point for a more systematic approach to examine the 'predictive value' of single resilience factors. In the revised version of our manuscript, we more extensively discuss the limitations of our approach and hope you find this a more balanced view on our method. Please see below for a point-by-point response to your comments.

2. Table 1 (pag. 10) provides an overview of the criteria used to rate the evidence levels of the outcomes of include studies. The table shows that, as long as a resilience factor is significantly associated with resilient outcomes, the rating is positive (+ to +++): studies without control or other variables were rated with a +, studies with sociodemographic control variables with ++, and studies with sociodemographic control variables and other resilient factors with +++. The main and major problem of this strategy is that it tells the reader little to nothing about the relevance of the regarding resilience factors: significance level is a very poor and unacceptable measure for the evidence level and should be replaced with well-established effect size measures across studies (see meta-analysis). Trivial or minimal and thereby irrelevant associations can easily be significant when the sample is large enough (the social science literature is full of such associations). It is therefore impossible to offer critical and helpful comments on large parts of the results (and discussion) section that were primary based on the evidence ratings (the unclear distinction between resilience and sociodemographic factor is a minor problem in this perspective). In the results section the authors also provided an overview of factors that were more or less examined in the included studied, but I do not know if such an overview is of interest for the readers because the authors excluded all studies not using GGM or comparable methods (see further below).

Response: Many thanks for this critical comment. We agree with you that the use of statistical significance can be problematic (even though most of the evidence available from primary studies still uses this approach). Also, for our initial submission, we have discussed to amend our rating by information derived from effect sizes. In the first place, we have decided against such an approach as the synthesis of effect sizes derived from different regression models has been criticized within the literature on the use of Odds Ratios for meta-analyses (Chang and Hoaglin, 2017; Norton et al., 2018).

This is also our rationale for not performing meta-analyses. We totally agree with you that meta-analyses are superior to narrative syntheses or ordinal rating schemes. As our research group has a

strong focus on quantitative evidence synthesis, we would have favored such an approach above the use of an ordinal rating scheme. However, based on our literature review and discussions with statisticians, we believe that it is not appropriate when models include a highly heterogeneous number of control variables (1 to 43 in our rating scheme), which are also highly different in nature ranging from sociodemographic variables, indicators of stressor exposure to other resilience factors. The relevance of variables included for modeling is also supported by our analyses showing that evidence ratings were less favorable when more variables were included, and resilience factors were controlled for other resilience factors (see p. 28-29 in the manuscript).

Rethinking our initial decision against the use of effect sizes for synthesis in general, we aimed at developing a revised rating scheme combining both – evidence from statistical tests and evidence derived from effect sizes. For this purpose, we followed the effect size categories developed by Cohen (1988; i.e., very small, small, medium, large), which were adapted to ORs based on standard recommendations (Mood, 2010). Thereby, our ratings now provide a more nuanced view. Within our revised Results and Discussion sections, we critically reflect on these effect sizes and discuss the rationale for our approach along with its limitations.

We agree with you that a quantitative estimate per resilience factor would provide a superior summary of evidence, at the same time, we believe that the current state of literature does not allow to derive such an estimate. Thus, we are confident that our approach provides a systematic summary of evidence that is not yet available for resilience factors and thereby significantly adds to the current state of knowledge in the field.

Relevant changes in the manuscript (p. 12):

For this systematic review, we aimed at providing a qualitative synthesis of findings amended by insights from additional quantitative analyses. Although we had estimates of odds ratios (ORs) or comparable regression coefficients from most of the included primary studies, these coefficients were controlled for highly heterogeneous variables (i.e., the number and nature of control variables varied substantially between studies). This prevented the use of standard meta-analysis for synthesis^{79,80}. Thus, all quantitative analyses are based on non-parametric statistical tests that can only be viewed as an add-on to our qualitative summary and need further replication using standard meta-analyses based on more homogeneous primary studies.

One minor remark on the effect of sample sizes: We agree with you that within the largest samples even very small and negligible effects will reach statistical significance. We have aimed at examining this effect by studying the association of ‘sample size used for modeling’ with evidence ratings based on our newly developed rating scheme. We found no significant link between the sample size used for modeling and evidence rating. If favorable ratings would have been largely driven by large sample, we would have expected to find such a significant link.

Relevant changes in the manuscript (p. 28-29):

There was no statistically significant evidence for an association between the total sample size at baseline and evidence ratings, $r_s = .02$, 95% CI [-.11, .13], $p = .749$. Evidence ratings were less favorable when a larger number of variables was included in logistic regression models, $r_s = -.22$, 95% CI [-.33, -.10], $p < .001$, and more resilience factors were simultaneously examined, $r_s = -.25$, 95% CI [-.36, -.13], $p < .001$.

With respect to the categorization of resilience factors: We agree with you that some of those factors referred to as resilience factors in our review may also be viewed as sociodemographic factors. In the revised version of our manuscript, we have added our rationale for differentiating resilience factors from sociodemographic variables. For the purpose of our review, we use the term ‘resilience factors’ for those modifiable factors that can be influenced by individual and/or systemic

resilience-promoting intervention or factors which may represent proxies of well-established resilience factors (e.g., family status as indicator of support). While (rather systemic) interventions might be able to promote education or increase income within societies, gender or age as non-modifiable factors cannot be subject to an intervention. At the most, interventions might be offered to specific age groups or genders, but those factors are (rather) fixed. This information is now provided in the Methods section.

Relevant changes in the manuscript (p. 10):

The classification of resilience factors was based on previous reviews^{16,17,20,47,48} in the field. However, as there is no finite list of resilience factors, we also included all kinds of factors that were discussed as potentially health- or resilience-promoting by primary study authors. **Moreover, some factors (e.g., education, income, family status or socioeconomic status) could either be classified as sociodemographic characteristic or resilience factor. In these cases, variables were included as resilience factors when they were either potentially modifiable by individual or systemic interventions¹³ (e.g., education, income) or might provide a proxy measure of rather well-established resilience factors (e.g., family status or living with a partner as indicators of available support).**

Other problems I encountered were:

3. The included studies examined trajectories of mental health by means of GMM (or comparable methodological approaches) aiming at identifying different patterns of mental health over time) and investigated individual, social and/or societal resilience factors as their predictors (i.e., as an independent variable in a regression analysis). Importantly, Table 2 shows that included studies differed strongly in timing of the assessments: part of the studies included pre-event/societal change assessments while other studies started one or more years of the event/societal changes and/or differed in timing of subsequent surveys. Part of the studies conducted the last assessment within 1-2 years after the event/societal changes, while other conducted the last assessment several years later. In other words, totally different study designs were applied in the included studies. How these important differences were accounted for in the analyses (including weighing of evidence levels) is not explained nor clarified, as if timing/time lags do not matter. I tried to understand it, but eventually did not make sense to me.

Response: Many thanks for highlighting this. We agree with you that there is substantial between-study heterogeneity in our review. One of the aspects contributing to between-study differences are aspects of timing (i.e., availability of pre-stressor data, time lags between stressor and data collection, length of post-stressor follow-up, total length of data collection). These challenges were also present in previous reviews summarizing trajectory-based resilience research (Galatzer-Levy et al., 2018; Infurna and Luthar, 2018; Lai et al., 2017; Schäfer et al., 2022). However, none of these reviews found effects of study design on prevalence estimates (e.g., based on a moderator analysis comparing studies with and without pre-stressor data). We agree with you that these aspects have to be further examined based on large-scale databases, which also motivated our ongoing evidence synthesis on trajectory-based resilience research (Thomas et al., 2023).

Within the current project, we aimed at addressing these issues by performing sensitivity analyses. In the revised version of your manuscript, we examined i) the time interval between the last pre-stressor assessment and occurrence of the stressor; ii) the time interval between stressor exposure and the first post-stressor assessment, and iii) the time interval between stressor exposure and last assessment. For none of those factors, we found evidence for a significant association with evidence ratings.

However, we cannot exclude that between-study heterogeneity in timing has introduced biases in our analyses. This limitation is now reported in the Limitation section of the review together with a

discussion of potential dynamics of the importance of specific resilience factors over time, which may not be adequately captured by our analyses. More complex primary studies might be more suitable to provide detailed insights into these aspects.

Relevant changes in the manuscript (p. 15-16, p. 37-39):

In our sensitivity analyses, **we examined the importance of study quality and timing**. For this purpose, we used Spearman's rank correlations and examined the association between study quality ratings and evidence ratings with a significant correlation coefficient suggesting a relevant impact of study quality. **Using the same approach, we examined whether there was a link between evidence ratings and i) the time interval between the last pre-stressor assessment and occurrence of the stressor; ii) the time interval between stressor exposure and the first post-stressor assessment, and iii) the time interval between stressor exposure and last assessment.**

[...]

Also, timing of post-stressor assessments varied substantially between studies, with some studies' assessments starting in the first hours after exposure and others after 3.3 years¹¹⁵. **Also the number of post-stressor assessments (from 3 to 27 assessments) and the length of follow-up varied substantially from 1.7 months to 14 years, which might also impact on the relative importance of resilience factors⁵⁵.** Another limitation of our review is missing **pre-stressor** data with only 24% of the studies including pre-stressor assessments. **Such** studies are particularly challenging in face of societal stressors, which often have a sudden and unforeseen onset such as terrorist attacks or the COVID-19 pandemic, preventing the collection of pre-stressor data. **At the same time, those studies are needed as a lack of pre-stressor assessment may result in underestimated prevalence rates for recovery responses (as initial increases of distress have been missed)¹⁶.**

[...]

Moreover, such studies may also make use of advanced methods for predictor selection (e.g., machine learning approaches¹¹⁸, LASSO regression²⁴), which provide knowledge on the relative importance of resilience factors moving beyond statistical significance⁸¹. **Those studies may also allow to identify dynamics of the importance of resilience factors over time, which are not captured by our review. For example, in a recent study¹⁰⁷, the importance of perceived social support was found to vary in the first year after stressor exposure and also between different sources of social support.**

4. In addition, the type of events/societal changes the included studies focused on differ strongly. The majority focused on the COVID-19 pandemic and others on specific events like the 9/11 attacks. Please clarify why you believe that the outcomes if these studies can be combined and not separated by type of event. I could not discern much similarities between (for example) these two type of events.

Response: Many thanks for this comment. We agree with you that societal challenges are still a non-homogeneous category of events/stressors. This is a challenge relevant to most reviews in resilience research that are in many cases not related to a specific type of stressor (Dewar et al., 2020; Galatzer-Levy et al., 2018; Garfin et al., 2018; Gerber et al., 2021; Infurna and Luthar, 2018). From our point of view, the range of stressors included in this review is relatively small compared to similar work in the field. For the in- and exclusion of stressors we build on an up-to-date list of stressors used in the field of public health (Leppold et al., 2022). All stressors considered in this review occur at a societal rather than an individual level, that is, they affect the total population (even to a different extend), relatively synchronously and may have a long-lasting impact on individual lives and societies affected by those stressors. For example, both the COVID-19 pandemic

as well as 9/11 attacks have affected societies over longer periods and have resulted in substantial changes for the lives of many (e.g., by major changes in policies), while individual-level stressors like serious health events or divorces do not result in such larger-scale changes. Moreover, those types of stressors may have scaling effects, that is, they result in a potential series of stressful life events caused or amplified by the stressors (e.g., loss of loved ones, job insecurities/loss, burden caused by childcare during COVID). However, we agree with you that these similarities were not sufficiently addressed in the previous version of our manuscript. We have added this information for the revised version and included the between-stressor heterogeneity to our Limitations section. Especially, we clearly state that the current evidence base does not allow for sufficiently powered and sound between-stressor comparisons, which would require a larger number of primary studies for specific stressors. Only based on such studies, we can or cannot conclude that there are no between-stressor differences for the importance of specific resilience factors. This is now clearly stated in our manuscript.

Relevant changes in the manuscript (p. 3, p. 36-37):

Within the last years, many societies were exposed to multiple stressors and crises such as the COVID-19 pandemic, economic crises, wars and armed conflicts, or natural disasters¹. **Beyond differences between those stressors, they share relevant similarities as they affect a large number of people relatively synchronously and have potentially long-lasting consequences for societies², leading to increased stress for many individuals^{3,4}.**

[...]

These studies will also provide empirical insights into whether the selection of stressors used for this review² represents a sufficiently homogeneous subgroup or needs further refinement. Such a typology of stressors moving beyond traumatic stress^{113,114} is urgently needed and may also help to classify stressors along important dimensions (e.g., intensity, predictability, controllability, novelty, duration³⁹).

5. I understand that the authors used the term resilience as an outcome or label: the maintenance or fast recovery of mental health during or after stressor exposure. In many studies resilience is, in brief, defined as a personal capacity “to bounce back” during or after stressful events or circumstances. It could help the reader if the authors explained why they choose the “outcome” definition.

Response: Thank you for highlighting that. We agree with you that the “personal ability to bounce back” is among the most commonly used definitions of resilience. From our point of view, which is in line with recent theoretical discussions in the field (Bonanno et al., 2023; Kalisch et al., 2017; Southwick et al., 2014), this personality-based and often trait-like definition comes with major shortcomings. Among many, this definition implies that resilience is a rather stable trait with little dynamics between responses to different stressors or over the lifespan. However, there is robust evidence underscoring substantial dynamics reflected in between-stressor and between-outcome differences and relevant changes over the lifespan (Infurna and Luthar, 2017; Masten, 2019). In line with recent theoretical claims (Chmitorz et al., 2018), we have based our review on a multifaceted view of resilience differentiating between resilience factors, resilience mechanisms, and resilient outcomes (Kalisch et al., 2015). However, we agree with you that it might be good to discuss this approach in the context of the well-known definition of “bouncing back”. This has now been included in the revised version of our manuscript.

Relevant changes in the manuscript (p. 3):

While resilience has often been viewed as the personal capacity to bounce back after exposure to stress⁹, recent approaches in resilience research conceptualize resilience as an outcome, that is, favorable adaptation in face of stress^{10,11}. More precisely, resilience as an outcome can be defined as the maintenance or fast recovery of mental health during or after stressor exposure¹¹. **Within this framework**, so-called resilience factors protect individuals from potentially negative effects of stressors and increase the likelihood of resilient responses^{12,13}.

6. *In addition, if no (non-retrospective) pre-event mental health data are available in the included studies, what do post-event trajectories actually say? For instance, if people already suffered from pre-existing severe mental health problems and belong to the so-called chronic trajectory, are they then not resilient? Please explain.*

Response: We also agree with you that trajectory-based research into resilience is not without criticism (Infurna and Luthar, 2018, 2016; Schäfer et al., 2022). Among the most relevant shortcomings of trajectory-based resilience research is the lack of pre-stressor data, which might - among other problems - lead to recovery trajectories being overlooked. A full review on trajectory-based resilience research is beyond the scope of our current work, which focuses on resilience factors as the predictors of these trajectories. However, we agree with you that these shortcomings are important to the appraisal of our results. Thus, we have added a critical appraisal of trajectory-based resilience research including problems resulting from study design (e.g., no pre-stressor data is available; Kalisch et al., 2015; Schäfer et al., 2022) and methodological decisions (e.g., overrating of resilient responses due to the use of very restrictive models; Infurna and Luthar, 2018, 2016). We suggest to add this as a box as we believe that it is not the focus of our work, but can be used to provide the reader with important background information. Moreover, we discuss the implications of these limitations for our review findings. Specifically, we discuss that the lack of pre-stressor data might result in identifying resilience factors as important predictors that are rather correlates of stable between-person differences in mental health and not specifically relevant to responses to stress. Moreover, an overidentification of resilient responses might introduce additional error and may result in a lack of power for the identification of resilience factors that differentiate resilient from other responses. This is now addressed in our Limitations section.

See comment #10 for our response to problems caused by non-representativeness of primary studies.

Relevant changes in the manuscript (see Box 1, p.5, p. 35):

However, this view has been challenged by the work of Infurna and Luthar^{40,41} who aimed to replicate previous findings in the field and found the results to largely depend on modeling decisions. In a later review paper⁴², they claimed that especially the use of highly restrictive models resulted in high prevalence estimates for resilience. Those restrictive models assume that variances are homogeneous across trajectories and slopes are equal within one trajectory. When using less constrained models, prevalence estimates for resilient responses were decreasing. In response, Galatzer-Levy et al.⁴³ stated that the use of less constrained models also reduces their exploratory value.

What are the consequences of this ongoing debate for the current review? It is beyond the scope of this review to present a comprehensive overview on trajectory-based resilience research. In our review, we use individual class assignments as outcomes and examine the predictive value of resilience factors for the (most likely) class membership, with many studies using regression methods also accounting for uncertainty in class assignments³⁷. Thus, there might be some imprecision in our findings resulting from biases in trajectory-based resilience research, yet it is unclear whether this induces rather random error and may thus result in a lack of power or biases

our results in a specific direction (i.e., overrating or underrating the relevance of specific resilience factors).

[...]

However, the use of GMM is a methodological approach which is not without criticism^{42,109} (see also Box 1), with models being criticized for artificially producing inflated prevalence rates for resilient responses by being highly constrained^{40,109}. Moreover, a lack of pre-stressor data might also result in inflated prevalence estimates for resilient responses as initial increases in distress might be missed resulting in recovery responses being mis-labeled as resilience trajectories. Yet, previous reviews^{16,29} did not provide evidence for such a bias. Inflated prevalence rates for resilient responses may also impact on our evidence ratings for resilience factors and might increase random error, which could have induced bias in both directions – important factors might be missed, or the importance of resilience factors might be overestimated.

7. In the trajectories figure on page 44 a sub-class of the so-called resilience trajectory is presented where mental health increase after the event/societal changes but decrease (so it seems) relatively fast. Please explain after how many days/weeks/month initial mental health problems must decrease to define this group as resilient to be able to compare studies.

Response: We also agree that in some cases the differentiation between resilience and recovery might be debatable. From a theoretical point of view, a quick regain of mental health after stressor exposure is also conceptualized as ‘resilience’ (Kalisch et al., 2015). This reflects the idea that major stressors do not leave people entirely unaffected. However, a concrete timeline for these fast recovery processes is still missing, which has also been criticized previously (Schäfer et al., 2022). Empirically, those trajectories are very rare and might not be easy to identify by means of current longitudinal research as those might require assessments on a weekly basis that are often missing. Therefore, we decided to keep this type of trajectory in the figure presenting our theoretical idea as we do not want to claim that any kind of mental response to stress is pathological per se, but we have chosen a lighter color in the revised version and explain this decision in the caption of the figure.

Relevant changes in the manuscript (p. 4):

Figure 1. Illustration of the link between resilience factors and resilience outcomes mediated via higher-level resilience mechanisms. This idea is based on recent ideas in resilience research^{11,20,29} and has been adapted for the multilevel resilience factor approach of this review. Note that also (neuro)biological and (epi)genetic factors are discussed as individual resilience factors³⁰, however, those are not focus of this review. *In this figure, we present a trajectory of fast recovery after stressor exposure in light red, which is also labeled as ‘resilience’.* This reflects the idea that not any kind of mental response to stress is pathological per se²⁰. However, we acknowledge the fact that such trajectories of very fast recovery of mental health have rarely been identified in primary studies, which might also reflect problems of timing as such responses could only be captured by high-frequency assessments³¹.

8. But more importantly, the authors would help the reader much more by explaining the advantage of assessing factors associated with trajectories as assessed with GMM above assessing factors associated with the development of mental health problems at different moments after the event/societal changes in the aftermath (that certainly enables classifications of trajectories of for instance PTSD too). Do they not inform us about individual, social; and social risk factor at different moment after the event/societal changes? I do not think so.

Response: Many thanks for this critical comment. We agree with you that such an approach might also be of interest. For the purpose of our review and in line with our prospective preregistration (<https://osf.io/9xwyu/>), we decided to focus on trajectory-based resilience research. This has been done as it is probably the most common approach in the field (Bryant, 2021; Galatzer-Levy et al., 2018) when resilience is defined as an outcome. Moreover, we totally agree with you that there is a lot of heterogeneity between studies in the field and we decided to stick to a specific methodological approach to limit between-study differences in our review. We also believe that studies assessing the association of mental health and resilience factors at multiple time points after trauma are also at risk for capturing rather cross-sectional associations than predictors of dynamics over time.

However, we agree with you that findings from studies using other (and more complex) methodological approaches to model bidirectional dynamics over time might result in different conclusions. Therefore, we clearly state that there is a need for studies assessing the dynamics between resilience factors and post-stressor changes in mental health using more complex and dynamic approaches (Meuleman et al., 2024; Schäfer et al., 2022; Sippel et al., 2024). At the same time, we believe that these studies answer different research questions and should thus be summarized in other systematic reviews more suitable to synthesize these findings.

Relevant changes in the manuscript (p. 38):

A larger number of high-quality international panels will allow for integrating data into a large-scale dataset suitable for individual participant meta-analysis¹⁷, which will help to shed light on participant-level modulators (e.g., age and gender). Our review suggested that the importance of resilience factors might be larger for women and younger populations, however, our database did not allow for an in-depth analysis on gender- and age-related differences for resilience factor levels or single factors. Moreover, such studies may also make use of advanced methods for predictor selection (e.g., machine learning approaches¹⁸, LASSO regression²⁴), which provide knowledge on the relative importance of resilience factors moving beyond statistical significance⁸¹. **Those studies may also allow to identify dynamics of the importance of resilience factors over time, which are not captured by our review. For example, in a recent study¹⁰⁷, the importance of perceived social support was found to vary in the first year after stressor exposure and also between different sources of social support.**

9. GGM and latent class analyses are very interesting techniques but not without limitations. There is always classification error and studies differ in the amount of classification error. As far as I could see in the materials I received, this topic was (much to my surprise) ignored in the quality assessment of the included studies as well as in the evidence ratings.

Response: In line with your responses to your previous comments, we agree with you that GMMs are not without limitations. We address these limitations now more prominently in the manuscript (see Box 1). Moreover, based on your comment, we have decided to include two model-related items to our quality assessment. We did not do this in the first place as our review question is more about the subsequent regression analyses than about the modeling itself. However, we agree with you that problems resulting from poor (i.e., highly restrictive) modeling also impact on our findings (see our response to comment #3). In the revised version of our manuscript, we now rated whether models were highly restrictive in terms of fixing variances between trajectories and slopes within a trajectory and whether authors used a regression model accounting for uncertainty in class assignments. These two items are now included in our quality checklist and are available for each study included in our review.

Relevant changes in the manuscript (p. 6, p. 11-12, p. 18):

How did we address these methodological issues? Within the current review, we included an assessment of overall model restrictiveness (i.e., fixed variances and slopes) as part of our quality assessment, which was used for sensitivity analysis. As in previous reviews in the field¹⁶, reporting standards were low for modeling decisions, however, when modeling decisions were not or insufficiently reported, models were rated to be restrictive for a more conservative approach. Additionally, we included the quality of regression modeling, which is key to our review question, as part of our quality assessments, with high-quality studies considering uncertainty in class assignments for regression analysis³⁷.

[...]

Study quality was assessed by two team members as an indicator of risk of bias using a modified version of the Newcastle-Ottawa Scale (NOS^{74,75}), assessing bias from 1) selection, 2) comparability, 3) outcome assessment, 4) reporting of methodological details, and 5) quality of trajectory modeling (i.e., constraints of variances across classes and slopes within one class; see OSF project: <https://osf.io/9xwyu/>). Based on the number of items that could be assessed per study, we calculated an overall study quality rating, ranging from 0% to 100%.

[...]

Overall study quality was moderate (see Figure 3), with a median study quality rating of 66.67 ($M = 62.17$, $SD = 11.23$; range: 33.33–94.44). Main flaws across primary studies were found for quality of GMM (74.0% high risk), selection (14.0% high risk), followed by outcome assessment (8.0% high risk), and comparability (6.0% high risk). There were no differences with respect to study quality between studies that assessed either individual, social or societal resilience factors, or a mix of different level resilience factors, Kruskal-Wallis $H = 4.9$, $p = .129$, 95% CI [.127, .140].

10. With respect to the quality assessment, the authors wrote that “Only a small share (11 studies) was representative of the respective target population with most using convenience samples”. A well-known major problem with convenience samples is that we simply do not know whether the results are representative for a certain population or not. Please explain or clarify why it is justified to include studies with an unknown representativeness. In my personal view, such studies should be excluded because the quality rating is by definition (very) low.

Response: We also agree with you that representativeness is a very important aspect in research. However, we refrained from excluding all studies that use non-representative samples from our review. In case of our review, for many studies it is hard to tell what would be the target population that should be represented. This might be all people living in a specific country when we examine the consequences of the COVID-19 pandemic. However, for other disasters it is hard to tell whether the target population would be all people being exposed to a stressor or whether a severity of stressor exposure is essential to be considered as a part of the target population. The same applies to natural disasters, wars, and armed conflicts. Thus, we believe that addressing this problem by means of a sensitivity analysis and as a potential limitation is important but excluding all studies using non-representative populations would have been a too big loss of information.

Relevant changes in the manuscript (p. 29, p. 37):

There was no statistically significant evidence for differences between representative or non-representative samples, $U = 6397.50$, $p = .161$, 95% CI [.154, .168].

[...]

We examined by means of sensitivity analyses whether the availability of pre-stressor data impacted evidence ratings, without finding statistically significant evidence for differences between studies with and without pre-stressor data. Yet, we cannot exclude that these studies have introduced biases in our evidence ratings. Other problems might be caused by missing representativeness and low diversity of study samples. For example, we cannot exclude that those who were affected the most by stressors (e.g., minority groups¹¹⁶) were not sufficiently healthy to participate in the studies included in our review, which could have resulted in substantially biased findings.

11. As said above, I am very sorry to say that in my view the main research questions 4 and 5 are not really answered because of the chosen analytic strategy. At this moment, I cannot oversee the relevance of manuscript that answers the research question 1-3 only. Nevertheless, the authors created a large impressive database of studies. I hope that the authors continue with their important work and that my review is of help to a certain extent.

Response: Many thanks for your critical comments. We believe that those comments helped us to prepare an improved version of our manuscript. Specifically, we have revised our rating scheme and now no longer simply rely on statistical significance but also on effect sizes and use both aspects for our interpretation. Moreover, in the revised version of our manuscript, limitations of the GMM approach are addressed in greater detail and we have included two items on the quality of GMM in our quality assessment. We agree with you that our approach is not without problems. However, we believe that these problems should not prevent us from summarizing evidence on resilience factors. So far, most reviews simply rely on cross-sectional association of resilience factors and mental health. In many cases, reviews are based on populations that were not even exposed to significant stress (Gallagher et al., 2020). From our point of view, such evidence is much weaker than evidence that is available from the studies included in this review. There is a need for further studies using more homogeneous approaches that also allow for meta-analysis, however, we are confident that our review may help to pave the way for these studies and for more elaborated approaches to summarize more homogeneous studies in the future.

Reviewer #2:

1. This manuscript presents a systematic review examining the predictive validity of individual, social, and societal resilience factors in maintaining mental health trajectories during major societal crises. The topic is highly significant given recent events highlighting the need to understand public resilience. Key strengths of the review include the longitudinal, multi-level approach, examination of incremental validity, and focus on understudied societal factors. The review makes an important contribution in consolidating current evidence across crises and identifying critical gaps to guide future research. However, enhancements to the methodology, context, and implications would further strengthen the quality and impact of the findings.

Response: Many thanks for your time and effort dedicated to our work. We were pleased to read that you found our manuscript a valuable contribution to the existing knowledge on resilience factors. Based on your comments and the comments of Reviewer #1, we have made some substantial changes to our manuscript. Specifically, we now provide more information on potential multi-level resilience mechanisms, with very little knowledge being available on the social and societal level. Moreover, we discuss potential differences between types of stressors and provide more information on the categorization used for resilience factors in this review.

2. Provide more background on hypothesized resilience mechanisms that may mediate between factors and outcomes. Elaborate on what these mechanisms may be and how they operate at multiple systemic levels.

Response: Many thanks for this comment. We agree with you that potential higher-level resilience mechanisms are of great interest and that they probably look different at different levels of resilience factors. In the revised version of our manuscript, we have included more information on these potential mechanisms being aware of the fact that we have little insights into these mechanisms from the studies reviewed in our project. Thus, we have included some additional information in the Introduction section, but mostly focus on these mechanisms as a potential target of future research in the Discussion section. Especially at the social and, even more pronounced, at the societal level, there is a lot of conceptual work to do as theories on mechanisms at these levels are missing. Important insights might also derive from interdisciplinary work in the fields of resilience research, public health, security research, and sociology as ideas from other disciplines for example on transformation processes and resource availability might be a great inspiration for multi-level resilience mechanisms. This is now also stated in the manuscript.

Relevant changes in the manuscript (p. 3-4, p. 34):

For example, **at the individual level**, Kalisch et al.²⁰ proposed that many psychological resilience factors impact on resilient outcomes, with positive appraisal style (i.e., non-pessimistic, non-catastrophic, non-helpless types of appraisal¹⁶) being the key mediator. In a similar vein, Bonanno^{21,22} suggested regulatory flexibility as an overarching mechanism for resilient outcomes, with regulatory flexibility reflecting one's ability to modulate emotional experiences and the perceived ability to make use of different coping strategies depending on contextual demands and feedback. **So far, a small number of primary studies²³⁻²⁶ provides support for these ideas; however, evidence is still rare and comprehensive tests of more complex models also including associations with resilience factors are still missing. At the social and societal level, other resilience mechanisms come into play (e.g., decision making, use of resources, capacities for transformation)^{27,28}, however, those have rarely been examined in primary studies on individual mental responses to societal challenges.**

[...]

Our review identified research into social and societal resilience mechanisms as one of the most important evidence gaps. On a social level, one may think of mechanisms relevant to establishing and maintaining social relationships, which in turn lead to a general feeling of connectedness¹⁰⁵. On a societal level, the mobilization of resources (e.g., facilitating communication between stakeholders), self-regulation processes (e.g., capabilities to make decisions in times of crises) and capacities to transform societal systems (i.e., the ability to learn from previous challenges) might be viewed as potential resilience mechanisms²⁸. Future research will benefit from integrating knowledge from other fields examining societal adaptation processes (e.g., sociology, security research) and from adapting a multilevel and multisystemic perspective¹⁴, and might start with disentangling resilience factors and mechanisms at a conceptual level. Such conceptual clarity will help to examine the complex interplay between resilience factors and mechanisms at different levels.

3. Summarize differences/similarities in resilience patterns and factors across the societal crises, both from current evidence and informed speculation based on crisis characteristics.

Response: Thank you very much for this inspiring idea. We have now added information on differences and similarities of resilience patterns and resilience factors across multifaceted crises. Unfortunately, we have only little evidence for specific types of crises and our evidence base is biased by a large number of studies assessing the consequences of the COVID-19 pandemic (Schäfer et al., 2022). However, in our Discussion section we now discuss characteristics of crises that might be important to be examined in future studies or in larger projects. Specifically, we discuss that differences in crises may account for both differences in mental responses as well as differences in the importance of specific resilience factors relevant to adapt to these challenges.

Relevant changes in the manuscript (p. 36-37):

Thus, future studies need to examine whether our findings also hold for other stressors falling into the same larger category and whether resilience factors might only be beneficial for some types of stressors but not for others. *These studies will also provide empirical insights into whether the selection of stressors used for this review² represents a sufficiently homogeneous subgroup or needs further refinement. Such a typology of stressors moving beyond traumatic stress^{113,114} is urgently needed and may also help to classify stressors along important dimensions (e.g., intensity, predictability, controllability, novelty, duration³⁹).*

4. Discuss in more depth the policy, practice and crisis preparedness implications beyond informing resilience interventions. Particularly relevant are building societal resilience capabilities.

Response: Many thanks for this comment. In the revised version of our manuscript, we have added a section reporting on the implications of our findings for building resilience at different levels. At the same time, we clearly state that our findings should not be misinterpreted as showing a causal link between resilience factors and specific responses to stressors. Future studies using other designs would be needed to provide an evidence base for such causal links between specific factors and responses to stress over time. Such a causal link would represent an ideal foundation for the development of resilience interventions and capabilities.

Relevant changes in the manuscript (p. 39):

On the long run, **high-quality international panels** will inform the development, improvement, and evaluation of prevention and resilience programs. **So far, most resilience programs focus on the individual and aim at strengthening individual resilience factors^{13,17}, implicitly transferring responsibility for resilience to individuals rather than groups or societies. Our multilevel approach to resilience factors might have the potential to guide the development and evaluation of future programs as those may be designed to address resilience factors at multiple levels¹¹⁹. Constant multidimensional measurement of central cross-situational resilience factors on a national level can help guide public promotion or interventions efforts. At the same time, program evaluations should move beyond measuring changes in mental health and psychological resources^{17,120}. Future studies should also derive recommendations for so far neglected suitable outcomes at the social and societal level¹²¹. Holistic programs might employ a multilevel approach to all components of the process: resilience factors, resilience mechanisms and outcomes. Initial initiatives in this direction were outlined and discussed, particularly during the COVID-19 pandemic as a global stressor, in relation to crises preparedness and responsiveness at various levels (e.g., individual and community resilience)¹²². The studies summarized in this review could be a useful extension for possible starting points for such multilevel evaluations where event-related resilience interventions could be examined as predictors of post-stressor changes in multidimensional outcomes.**

5. Enhance methodological detail and clarity throughout, particularly around data extraction, analysis procedures, and growth mixture modeling rationale.

Response: Many thanks for this comment. Based on your comment and comments of Reviewer #1, we provide more details on our methodological approach. Specifically, we provide full information on our rating scheme and a more explicit rationale for focusing on growth mixture modeling to allow for greater between-study comparability (see Box 1). Moreover, we have also added more information on the limitations arising from these decisions along with potential implications for our results. More details on all data, code and materials are available from the OSF project associated with this review (<https://osf.io/9xwyu/>). We hope this increases the transparency of our approach and allows for full reproducibility of our findings.

6. Acknowledge limitations with the longitudinal observational design, sample diversity, and consistency of data quality over time.

Response: In the revised version of our manuscript, we have included these aspects in our Limitations section.

Relevant changes in the manuscript (p. 18, p. 37):

Attrition was insufficiently reported in many studies, but attrition rates were high for most studies (i.e., up to 99%⁹⁰), **indicating decreasing data quality over time.**

[...]

Other problems might be caused by missing representativeness and low diversity of study samples. For example, we cannot exclude that those who were affected the most by stressors (e.g., minority groups¹¹⁶) were not sufficiently healthy to participate in the studies included in our review, which could have resulted in substantially biased findings. Consequently, we cannot conclude that those resilience factors identified as important resources are equally important to all people exposed to a specific stressor in OECD member states.

7. Refine categorization of resilience factors, clarify their derivation, and discuss their theoretical grounding.

Response: Based on your comment, we decided to provide more information on that in our manuscript. The broader resilience factor categories were based on previous work of our group in the field (Kunzler et al., 2020a,b; Schäfer et al., 2022), where we derived categories based on theoretical considerations (e.g., factors regularly studied in the field of emotion regulation) and based on commonly used measures of the respective factors (e.g., coping measures). We agree that there are potential other classifications, however, we believe that broader categories help the reader to draw conclusions, while a simple long list of single factors is harder to interpret. We have added this information to our figure and also point to potential other classifications and overlaps between categories (e.g., cognitive emotion regulation and coping strategies).

Relevant changes in the manuscript (p. 23):

Broader categories of resilience factors (e.g., coping strategies, control beliefs) were defined based on previous work on resilience factors^{16,17,47,48} and well-established assessments^{91,92}; however, categories are not distinct, e.g., positive reframing might also be seen as a type of coping strategy.

Response: With respect to the assignment of resilience factors to specific levels, we have added the respective information on the rating process. For each factor, we coded the level at the stage of initial data extraction. This coding was checked by a second reviewer. All unclear cases were then discussed within the larger team of all review authors. However, as for all kind of ratings processes, we agree that the classification for some factors might have been different for other teams. This is now acknowledged within the Limitations section.

Relevant changes in the manuscript (p. 11):

Resilience factors were classified as either representing individual, social or societal resources by one reviewer, with individual resources being psychological dispositions, beliefs, or capabilities. Social factors were resources that were perceived or available in one's nearer social environment (e.g., family, friends), while societal factors were resources in the wider environment or the whole society (e.g., trust in authorities, legal protection; see SM4 for details on this classification). The decision on resilience factor level was checked by a second reviewer, with all disagreements being discussed and solved in the review team.

8. Explore the potential for more complex statistical modeling of the relationships between resilience factors and mental health trajectories.

Response: Many thanks for this comment. We agree with you that it would be great to have a more elaborated summary of evidence on resilience factors, ideally a meta-analysis allowing to derive effect estimates per factor. As our research group has a focus on evidence synthesis and meta-analysis, we put a lot of thoughts on the quantitative synthesis in case of our review. Unfortunately, we believe that a meta-analysis is not suitable. This decision is based on the nature of effects that can be extracted from primary studies. These derive from (multinomial) logistic regression analysis and are mostly based on Odds Ratios (ORs). In general, one may perform a meta-analysis of ORs. However, from our point of view as well as based on discussions with statisticians and a comprehensive literature review (Chang and Hoaglin, 2017; Norton et al., 2018), such a summary would not be suitable in our case as ORs within primary studies are controlled for a highly diverse set of potential confounders that vary in number and nature. Thus, we aimed at deriving as much quantitative information on evidence as possible based on the substantial between-study

heterogeneity. Future studies with smaller between-study differences may allow for meta-analysis and to derive effect estimates per resilience factor. This is now also stated in the future research section. Moreover, we point to the potential of more complex modeling approaches that can be employed within primary studies that should also examine the temporal dynamics and potentially reciprocal interplay between resilience factors and mental health over time.

Relevant changes in the manuscript (p. 12, p. 35):

For this systematic review, we aimed at providing a qualitative synthesis of findings amended by insights from additional quantitative analyses. Although we had estimates of odds ratios (ORs) or comparable regression coefficients from most of the included primary studies, these coefficients were controlled for highly heterogeneous variables (i.e., the number and nature of control variables varied substantially between studies). This prevented the use of standard meta-analysis for synthesis^{79,80}. Thus, all quantitative analyses are based on non-parametric statistical tests that can only be viewed as an add-on to our qualitative summary and need further replication using standard meta-analyses based on more homogeneous primary studies.

[...]

Thus, our findings should not be misinterpreted as evidence for a core set of resilience factors and mechanisms that should be targeted in resilience interventions. Such intervention targets should be derived from studies using more complex designs to investigate the interplay and potentially causal links between resilience factors/mechanisms and mental responses over time^{107,108}.

9. Addressing these areas will further enhance the quality, rigor and potential impact of this timely review synthesizing the current understanding and future directions of resilience research across systemic levels and societal shocks.

Response: Many thanks for your critical and inspiring comments. They were very helpful to prepare a revised version of our manuscript, which hopefully addresses your concerns.

23rd Jul 24

Dear Professor Schäfer,

Your manuscript titled "Individual, social and societal resilience factors in the face of societal challenges and crises: A systematic review of adult populations from OECD member states" has now been seen by our reviewers, whose comments appear below. In light of their advice I am delighted to say that we are happy, in principle, to publish a suitably revised version in *Communications Psychology*.

We therefore invite you to revise your paper one last time to address the remaining concerns of our reviewers and a list of editorial requests. At the same time we ask that you edit your manuscript to comply with our format requirements and to maximise the accessibility and therefore the impact of your work.

EDITORIAL REQUESTS:

SUBMISSION INFORMATION:

OPEN ACCESS:

* **DATA AVAILABILITY:**

[link redacted]

Best regards,

Jennifer Bellingtier

Jennifer Bellingtier, PhD

Senior Editor

Communications Psychology

REVIEWERS' EXPERTISE:

Reviewer #1 Stress, resilience, societal health and well-being

Reviewer #2 Stress, resilience, societal health and well-being

REVIEWERS' COMMENTS:

Reviewer #1 (Remarks to the Author):

I have read the revised manuscript with great pleasure and want to compliment the authors for all the work, clarifications and improvements. My main comments were addressed very well and, while I do not always agree with their arguments (such as that I still believe that studies based on weird convenience samples should be excluded), I honestly believe that the manuscript is almost ready to be accepted and published (In my view, authors and reviewers do not always have to agree on all arguments, decisions, formulations etc. as long as they are not misleading or simply false).

I only have a few small comments:

- The authors wrote “Only 12 studies (24.0%) included pre-stressor data”. Readers might be especially interested in those studies and I therefore ask the authors to add the reference numbers in the text.

- there are 3 summary tables showing which of the assessed factors are (not/limited) of relevance. I believe it would help the reader if the authors add the number (between brackets) of studies assessing a specific factor. For instance in the first table Individual Resilience Factors, instead of Overall emotion regulation, Overall emotion regulation (5).

- At the end the authors concluded “There is no single resilience factor outshining all other factors”. I believe that the authors are jumping into a conclusion here” (similar to sections in the discussion on this topic). I assume that authors meant “Of the assessed resilience factors in this review, there is no factor outshining all other factors”. The main reason why I mention this issue is because the authors did not examine the extent to which pre-event mental health problems are predictive of (trajectories of) post-event mental health problems, while research in many areas of the social sciences consistently show that the strongest predictor of a (post-event) variable is the same variable assessed at an earlier stage. Of special interest here are the reviews of DiGangi et al. (2013), Danese et al. (2017) and Scheeringa (2021). In other words, the factor (pre-event mental health problems) that most likely is the strongest (and strong) predictor was not examined (due to the analytic strategy of the authors and their focus on trajectories). I believe that this limitation should be added/clarified in the limitations sections and in the final conclusion section much better.

Reviewer #2 (Remarks to the Author):

The revised manuscript has been substantially improved in response to the reviewer comments. The introduction now provides a more comprehensive background on hypothesized resilience mechanisms at multiple systemic levels, enhancing the conceptual framework of the study.

The authors have expanded their discussion to summarize differences and similarities in resilience patterns and factors across societal crises, both from current evidence and informed speculation based on crisis characteristics. This addition enhances the generalizability and potential implications of the findings.

The discussion section now includes a more in-depth exploration of the policy, practice, and crisis preparedness implications, particularly in building societal resilience capabilities. This strengthens the translational impact of the research.

The methodological details have been enhanced, providing greater clarity around data extraction, analysis procedures, and the rationale for focusing on growth mixture modeling. The authors have also acknowledged additional limitations related to the longitudinal observational design, sample diversity, and consistency of data quality over time.

The categorization of resilience factors has been refined, with clearer derivation and theoretical grounding. However, the potential for more complex statistical modeling of the relationships between resilience factors and mental health trajectories remains an area for future exploration.

There are, however, several lingering issues that I suggest addressing in the Discussion section:

While the authors have expanded on the potential for more complex statistical modeling, this remains an area for further exploration in future research to deepen our understanding of the dynamic relationships between resilience factors and mental health trajectories.

The conceptual framework of resilience mechanisms across multiple levels has been strengthened, but empirical evidence supporting these hypothesized mechanisms remains limited. Future studies should aim to directly test these multilevel resilience mechanisms.

The generalizability of the findings across different societal challenges could be further explored as more evidence accumulates, allowing for well-powered comparisons of resilience patterns and predictors across various types of stressors.

Overall, the authors have successfully addressed the majority of the reviewer comments, and the revised manuscript represents a contribution to our understanding of multilevel resilience factors in the face of societal challenges. The identified lingering issues primarily represent areas for future research to further advance this field of inquiry.

Reviewer #1:

I have read the revised manuscript with great pleasure and want to compliment the authors for all the work, clarifications and improvements. My main comments were addressed very well and, while I do not always agree with their arguments (such as that I still believe that studies based on weird convenience samples should be excluded), I honestly believe that the manuscript is almost ready to be accepted and published (In my view, authors and reviewers do not always have to agree on all arguments, decisions, formulations etc. as long as they are not misleading or simply false).

Response: Many thanks for your time dedicated to our manuscript. Your comments on our first version were extremely helpful for improving our rating scheme and the evidence synthesis. We agree with you that convenience samples are among the greatest issues in this field of research. In the revised version, we have further highlighted this short-coming and clearly state that future studies should recruit representative samples (with the definition of the population of interest still being a problem). Moreover, we have adapted our key figure and now include numbers of studies and effect estimates available for a specific resilience factors.

Relevant changes in the abstract (p. 2):

Most studies used non-representative convenience samples and effects were small when accounting for sociodemographic variables and other resilience factors.

Relevant changes in the manuscript (p. 30):

Other problems might be caused by missing representativeness, with most studies using convenience samples, and low diversity of study samples. In general, defining a target population for studies on consequences of stressor exposure is challenging. For some stressor types (e.g., environmental and natural disasters, terrorist attacks), one may ideally recruit a representative sample of those living in a respective area or being present during a specific event at a given time, for other stressors (e.g., pandemics) such approaches might be less suitable as between-individual heterogeneity in exposure levels is larger. Such challenges can be addressed by accounting for levels of exposure in statistical analyses²¹, yet this has not been sufficiently done in many studies. Especially for those studies using convenience samples, we cannot exclude that those who were affected the most by stressors (e.g., minority groups¹²⁴) were not sufficiently healthy to participate in the studies included in our review, which could have resulted in substantially biased findings.

I only have a few small comments:

1. The authors wrote "Only 12 studies (24.0%) included pre-stressor data". Readers might be especially interested in those studies and I therefore ask the authors to add the reference numbers in the text.

Response: Many thanks for this suggestion, we have added the respective references for studies including pre-stressor data.

Relevant changes in the manuscript (p. 15):

Only 12 studies (24.0%)⁹⁰⁻¹⁰¹ included pre-stressor data, while the remaining 38 studies started during exposure up to 40 months after stress exposure. Studies included 3 to 27 assessment waves ($M = 5.71$ waves, $SD = 4.28$), covering 2 months to 15 years post-stressor.

2. There are 3 summary tables showing which of the assessed factors are (not/limited) of relevance. I believe it would help the reader if the authors add the number (between brackets) of studies

assessing a specific factor. For instance in the first table Individual Resilience Factors, instead of Overall emotion regulation, Overall emotion regulation (5).

Response: Following your suggestion, we have added both the number of studies examining the respective resilience factor and the number of effect estimates available for the respective factor.

Relevant changes in the manuscript (p. 55–56 and Figure 4):

1. (Cognitive) Emotion regulation			
Overall emotion regulation (3/5)			
Emotional clarity (1/2)			
Peaceful disengagement (1/2)			

Note. Numbers in parentheses (e.g., overall emotion regulation [3/5]) indicate the number of studies and effect estimates available for the respective resilience factor (e.g., 3 studies were reporting on overall emotion regulation, with 5 effect estimates being available).

3. At the end the authors concluded “There is no single resilience factor outshining all other factors”. I believe that the authors are jumping into a conclusion here” (similar to sections in the discussion on this topic). I assume that authors meant “Of the assessed resilience factors in this review, there is no factor outshining all other factors”. The main reason why I mention this issue is because the authors did not examine the extent to which pre-event mental health problems are predictive of (trajectories of) post-event mental health problems, while research in many areas of the social sciences consistently show that the strongest predictor of a (post-event) variable is the same variable assessed at an earlier stage. Of special interest here are the reviews of DiGangi et al. (2013), Danese et al. (2017) and Scheeringa (2021). In other words, the factor (pre-event mental health problems) that most likely is the strongest (and strong) predictor was not examined (due to the analytic strategy of the authors and their focus on trajectories). I believe that this limitation should be added/clarified in the limitations sections and in the final conclusion section much better.

Response: Many thanks for this important remark. We totally agree with you that pre-event mental health is probably the most important predictor of post-event mental health/distress. From our point of view, this is not a resilience factor per se, but rather a previous assessment of the later outcome. In the revised version of our manuscript, we have added this information together with the respective references in our Introduction and Discussion section. Moreover, we clearly state in the Conclusions section that we are referring to the factors examined in our review.

Relevant changes in the manuscript (p. 5, p. 8, p. 32):

Among the most common approaches to study the importance of resilience factors is to examine their predictive value for resilience and, less common, recovery trajectories using a GMM approach, with some studies employing classical multinomial logistic regression analyses and others adopting a three-step approach accounting for uncertainty in class assignments³⁷. Some of those studies also account for the well-established predictive value of other variables such as pre-stressor mental health^{38,39} and previous stressor exposure⁴⁰.

[...]

The classification of resilience factors was based on previous reviews^{16,17,20,44,45} in the field and limited to multilevel psychosocial resources. Notably, other pre-stressor factors well known to

predict post-stressor mental health (e.g., pre-stressor mental health^{38,39}, lifetime stressor exposure⁴⁰) were not examined as resilience factors.

[...]

Among the resilience factors examined in our review, there was no single factor outshining all others.

Reviewer #2:

The revised manuscript has been substantially improved in response to the reviewer comments. The introduction now provides a more comprehensive background on hypothesized resilience mechanisms at multiple systemic levels, enhancing the conceptual framework of the study.

The authors have expanded their discussion to summarize differences and similarities in resilience patterns and factors across societal crises, both from current evidence and informed speculation based on crisis characteristics. This addition enhances the generalizability and potential implications of the findings.

The discussion section now includes a more in-depth exploration of the policy, practice, and crisis preparedness implications, particularly in building societal resilience capabilities. This strengthens the translational impact of the research.

The methodological details have been enhanced, providing greater clarity around data extraction, analysis procedures, and the rationale for focusing on growth mixture modeling. The authors have also acknowledged additional limitations related to the longitudinal observational design, sample diversity, and consistency of data quality over time.

The categorization of resilience factors has been refined, with clearer derivation and theoretical grounding. However, the potential for more complex statistical modeling of the relationships between resilience factors and mental health trajectories remains an area for future exploration.

Response: Many thanks for your time and effort dedicated to our manuscript. Your comments on the initial version of the manuscript and our revision were very helpful to improve the quality of our work and to deepen the discussion of our findings. Based on our comments, we have now added some aspects to our future research section and specifically highlight the so far not fully used potentials of more complex statistical modeling and more straightforward approaches to study multilevel resilience mechanisms.

There are, however, several lingering issues that I suggest addressing in the Discussion section:

1. While the authors have expanded on the potential for more complex statistical modeling, this remains an area for further exploration in future research to deepen our understanding of the dynamic relationships between resilience factors and mental health trajectories.

Response: Many thanks for this comment. We have further strengthened the lack of studies using more complex modeling approaches by including the call for such studies in our Conclusions section.

Relevant changes in the manuscript (p. 31):

Moreover, such studies may also make use of advanced methods for predictor selection (e.g., machine learning approaches¹²⁶, LASSO regression²⁵), which provide knowledge on the relative importance of resilience factors moving beyond statistical significance⁸¹. **By using more complex analyses**, those studies may also allow to identify dynamics of the importance of resilience factors over time, which are not captured by our review. For example, in a recent study¹¹³, the importance of perceived social support was found to vary in the first year after stressor exposure and also between different sources of social support.

2. The conceptual framework of resilience mechanisms across multiple levels has been strengthened, but empirical evidence supporting these hypothesized mechanisms remains limited. Future studies should aim to directly test these multilevel resilience mechanisms.

Response: We totally agree with you on the lack of studies directly testing those mechanisms.

Relevant changes in the manuscript (p. 33):

Future large-scale international studies on public mental health should include pre-stressor data, examine a larger number of resilience factors, should focus on social and societal resilience factors, invest in the **straight-forward** study of resilience mechanisms at multiple levels, **and employ more statistical modeling approaches suitable to capture complex temporal dynamics and between-factor interactions.**

The generalizability of the findings across different societal challenges could be further explored as more evidence accumulates, allowing for well-powered comparisons of resilience patterns and predictors across various types of stressors.

Overall, the authors have successfully addressed the majority of the reviewer comments, and the revised manuscript represents a contribution to our understanding of multilevel resilience factors in the face of societal challenges. The identified lingering issues primarily represent areas for future research to further advance this field of inquiry.

Response: Many thanks for your comments and inspiration for our future research section. We hope this review will provide a starting point for advances in the field.